# Contrastive Diffusion Alignment: Learning Structured Latents for Controllable Generation

Ruchi Sandilya [1]  Sumaira Perez [1]  Charles Lynch [1]  Lindsay Victoria [1]  Benjamin Zebley [1]
Derrick Matthew Buchanan [2]  Mahendra T. Bhati [2]  Nolan Williams [2]  Timothy J. Spellman [3]  Faith M. Gunning [1]
Conor Liston [1]  Logan Grosenick [1]

## Abstract

Diffusion models excel at generation, but their latent spaces are high dimensional and not explicitly organized for interpretation or control. We introduce ConDA (Contrastive Diffusion Alignment), a plug-and-play geometry layer that applies contrastive learning to pretrained diffusion latents using auxiliary variables (e.g., time, stimulation parameters, facial action units). ConDA learns a low-dimensional embedding whose directions align with underlying dynamical factors, consistent with recent contrastive learning results on structured and disentangled representations. In this embedding, simple nonlinear trajectories support smooth interpolation, extrapolation, and counterfactual editing while rendering remains in the original diffusion space. ConDA separates editing and rendering by lifting embedding trajectories back to diffusion latents with a neighborhood-preserving kNN decoder and is robust across inversion solvers. Across fluid dynamics, neural calcium imaging, therapeutic neurostimulation, facial expression dynamics, and monkey motor cortex activity, ConDA yields more interpretable and controllable latent structure than linear traversals and conditioning-based baselines, indicating that diffusion latents encode dynamics-relevant structure that can be exploited by an explicit contrastive geometry layer.

## 1. Introduction

Controlling the generation of complex system dynamics, such as fluid flows around an obstacle, neural responses to therapeutic stimulation, or evolving facial expressions, requires generative models whose latent spaces have a geometry that captures trajectories and allows movement along time and other system variables in a principled way. Despite the success of diffusion models in generative modeling, a fundamental limitation remains: their latent spaces are not explicitly organized to represent temporal or other auxiliary-variable–dependent structure, leaving no principled mechanism for traversing complex dynamics. Addressing this gap is crucial for applications where controllability is as important as fidelity.

Diffusion models provide unmatched fidelity, stable training, and reliable inversion (Ho et al., 2020; Dhariwal and Nichol, 2021; Rombach et al., 2022; Mokady et al., 2023). Yet their latents are not dynamics-aware: linear interpolations often produce implausible intermediate states (Wang and Golland, 2023; Hahm et al., 2024), and existing mechanisms that incorporate auxiliary information such as ControlNet or InstructPix2Pix (Zhang et al., 2023; Brooks et al., 2023) guide samples but do not yield consistent traversal directions across time or other variables. As shown in Sec. 5, this result in blurry or entangled transitions when interpolating fluid dynamics trajectories and inconsistent progression in neural recordings and facial expressions (Fig. 2).

Recent advances attempt to address this challenge. Geometry-preserving traversals (Hahm et al., 2024), physics-informed priors (Shu et al., 2023), and video diffusion models such as MCVD (Voleti et al., 2022), Imagen Video (Ho et al., 2022a), and Make-A-Video (Singer et al., 2023) improve interpolation or temporal realism, but they do not organize diffusion latents for controllable dynamics. The missing piece is a general recipe for dynamics-aware diffusion: a way to align latent geometry with system variables while preserving the generative fidelity of diffusion models.

We introduce ConDA, a flexible framework that addresses

---

[1] Department of Psychiatry, Weill Cornell Medicine, New York, NY, USA [2] Department of Psychiatry, Stanford University, Stanford, CA, USA [3] Department of Neuroscience, University of Connecticut School of Medicine, Farmington, CT, USA. Correspondence to: Ruchi Sandilya <sar4018@med.cornell.edu>, Logan Grosenick <log4002@med.cornell.edu>.

this limitation by separating *control* from *synthesis*. It organizes pretrained diffusion latents into a compact, contrastively-learned embedding space using auxiliary variables such as time, stimulation parameters, or expression intensities. Within this structured space, standard nonlinear operators (e.g., splines, finite differences, and LSTMs) enable smooth and interpretable trajectory traversal, while rendering is performed in the the original diffusion latent space to preserve generative fidelity. This separation of *editing* and *rendering* (see Fig. 1 and Sec. 3) transforms diffusion models into controllable generators of nonlinear spatiotemporal dynamics.

Across five different domains, ConDA consistently improves controllability while preserving fidelity. In fluid dynamics, ConDA achieves higher reconstruction quality (35.7 PSNR vs. 28.3 for linear baselines) and smoother flow-field interpolations (Fig. 2). In neural calcium imaging, it produces smoother temporal progression across activity states (Fig. 2). In neurostimulation, it captures class-dependent transitions that vary systematically with a given stimulation coil angle (Fig. 3). In facial expression dynamics, it preserves subject identity while enabling smooth traversal of expressions (Fig. 4). In monkey motor control, it organizes condition-dependent dynamics in delayed reaching tasks (Fig. 5). Together, these results demonstrate that contrastive organization reshapes high-dimensional diffusion latents into an interpretable, low-dimensional space in which simple nonlinear operators model dynamics effectively, improving classification and revealing neural trajectories aligned with velocity reaches.

**Our main contributions are:** 1. We propose ConDA, a framework for dynamics-aware diffusion that separates *editing* in a compact, contrastively learned embedding space from *rendering* in the original diffusion latent space. 2. We show that standard nonlinear traversal operators such as splines, finite differences, and LSTMs become effective when applied in this structured space, enabling smooth trajectory traversal beyond linear interpolation or direct conditioning. 3. We validate ConDA across five spatiotemporal domains (fluid dynamics, neurostimulation, neural calcium imaging, facial expression dynamics, and monkey motor control) demonstrating improved temporal consistency, controllable condition-dependent transitions, and interpretable latent trajectories aligned with underlying system dynamics.

## 2. Related work

**Diffusion.** Diffusion models are now the leading framework for high-fidelity image and video generation (Ho et al., 2020; Song et al., 2021; Dhariwal and Nichol, 2021; Rombach et al., 2022), offering photorealistic outputs, stable training, and practical deterministic inversion via DDIM (Mokady et al., 2023). Recent work has further improved diffusion

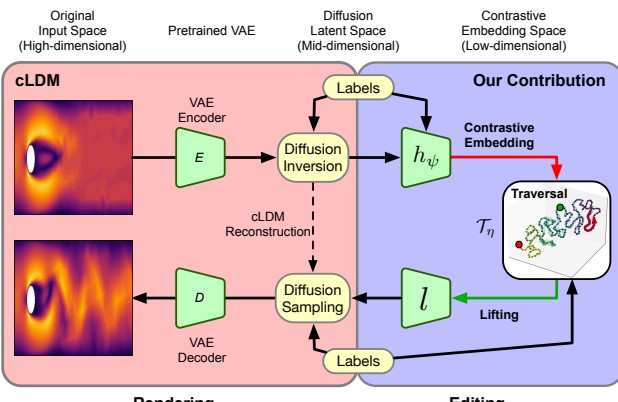

**Figure 1. Method overview of ConDA framework.** We encode a high-dimensional sequence of frames into a diffusion model feature latent trajectory via diffusion inversion (e.g., DDIM/Rex-RK4), then map these latents into a compact, contrastively learned embedding space that organizes local geometry to reflect nonlinear dynamics. An editing operator modifies the trajectory in the embedding space under new conditions (interpolation, local traversal, or class transfer). The edited embedding trajectory is lifted back to the feature latent space (via a learned or neighborhood-preserving kNN decoder) and decoded frame-by-frame by the diffusion sampler, yielding a new output sequence with controlled dynamics and preserved diversity and fidelity.

inversion by constructing algebraically reversible solvers such as Rex-RK4 (Blasingame and Liu, 2026). However, their latent spaces are not explicitly structured for dynamics: linear interpolations often yield unrealistic transitions (Wang and Golland, 2023; Hahm et al., 2024), and conditioning methods such as ControlNet or InstructPix2Pix guide edits without ensuring consistent traversal directions (Zhang et al., 2023; Brooks et al., 2023).

**Controllability.** Efforts to improve controllability include geometry-preserving traversals (Hahm et al., 2024), physics-informed priors (Shu et al., 2023), and video diffusion models (Ho et al., 2022b; Voleti et al., 2022; Singer et al., 2023) that emphasize realism over latent geometry. Related contrastive approaches such as NoiseCLR (Dalva and Yanardag, 2024) focus on discovering unsupervised, interpretable semantic directions for static image editing, while DRCT (Chen et al., 2024) leverages diffusion reconstruction and contrastive training for robust detection of diffusion-generated images. Time-series approaches such as Diffusion-TS (Yuan and Qiao, 2024), as well as label-efficient segmentation using diffusion (Baranchuk et al., 2022), remain task-specific and do not aim to learn a general-purpose, structured latent manifold for controllable spatiotemporal generation.

**GAN editing.** GANs revealed interpretable latent directions (Radford et al., 2015; Karras et al., 2020; Shen et al., 2020), inspiring disentanglement strategies such as Info-GAN (Chen et al., 2016) and $\beta$-VAE (Higgins et al., 2017). Yet GAN inversion is imperfect (Abdal et al., 2019; Xia

et al., 2022), latent directions are often linear, and editing typically remains prompt- or attention-based (Meng et al., 2022; Avrahami et al., 2022; Hertz et al., 2022) rather than organizing latent space into interpretable trajectories.

**Dynamics models.** Sequence models such as the Kalman filter (Kalman, 1960), SINDy (Brunton et al., 2016), RNNs e.g., LSTM/GRU (Hochreiter and Schmidhuber, 1997; Chung et al., 2014), structured SSMs (HiPPO, S4) (Gu et al., 2020; 2022; Smith et al., 2023), and neural ODE/SDE approaches (Rubanova et al., 2019) provide principled dynamics. Diffusion models have been combined with S4 and RNNs for forecasting and imputation (Alcaraz and Strodthoff, 2022; Rasul et al., 2021), while LDNS (Kapoor et al., 2024) focuses on learning low-dimensional latent dynamics with structured S4 models. We show that ConDA further improves interpretability by organizing LDNS diffusion latents into a dynamics-aware geometry.

**Disentanglement and contrastive learning.** Contrastive objectives structure latent embeddings by enforcing similarity and separation constraints (Oord et al., 2018; Chen et al., 2020; Poole et al., 2018; Wang and Isola, 2020), with successful applications in neuroscience (Schneider et al., 2023) and identifiability guarantees (Lyu and Fu, 2022; Cui et al., 2022). In contrast, diffusion disentanglement methods (Hahm et al., 2024) mainly address static factors. ConDA extends this line of work by incorporating auxiliary variables (e.g., time, stimulation, expressions) to align diffusion latents with spatiotemporal structure, enabling smooth nonlinear trajectory traversal.

**Summary.** Prior work advances interpolation, video realism, or disentanglement, but lacks a principled framework for organizing pretrained diffusion latents for nonlinear, interpretable control. ConDA fills this gap by (1) contrastively structuring embeddings with auxiliary variables, (2) enabling nonlinear traversal in this space, and (3) separating compact embedding edits from rendering in diffusion latent space. Our experiments show improved temporal consistency, smooth controllable trajectories, and faithful reconstructions across spatiotemporal data.

## 3. Method

### 3.1. Notation and Problem Statement

We consider controlled sequence generation for high-dimensional spatiotemporal data (e.g., videos of fluid dynamics, calcium imaging, neurostimulation E-fields, facial expression sequences, monkey electrophysiology). Each sequence $x_{1:S} = (x_1, \ldots, x_S)$ consists of a series of frames $x_s \in \mathcal{X}$ with associated per-frame auxiliary variables $y_s \in \mathcal{Y}$ (e.g., encoding time, neurostimulation coil angle, facial action units, etc.). Our goal is to generate a *target sequence* $\hat{x}_{j+1:j+n} = (\hat{x}_{j+1}, \ldots, \hat{x}_{j+n})$ frame-by-

frame under a user-specified auxiliary-variable trajectory $y'_{j+1:j+n}$ (e.g., a single target image is the special case $n = 1$). We study three evaluation settings: (i) *Interpolation*, where we regenerate intermediate frames under observed conditions $y'_{j+1:j+n} = y_{j+1:j+n}$ to assess the interpolation fidelity of the learned embedding space. (ii) *Held-out segment prediction (n-step-ahead)*, where given observed frames $(x_{1:j}, y_{1:j})$, we extrapolate the next $n$ frames by performing nonlinear traversal under target conditions $y'_{j+1:j+n} = y_{j+1:j+n}$. (iii) *Counterfactual editing*, where given two classes $(x_j^{(0)}, y_j^{(0)})$ and $(x_j^{(1)}, y_j^{(1)})$, we generate class-conditional interpolated trajectories to transfer from class 0 to class 1, with condition $y'_j = y_j^{(1)}$.

ConDA separates control from synthesis by introducing two latent spaces and a local-neighborhood lifting step. The pipeline proceeds in three stages. First, we train a conditional diffusion model (e.g., cLDM or pretrained IsoDiff) and apply diffusion inversion (e.g., DDIM or Rex-RK4) $g_\phi : \mathcal{X} \times \mathcal{Y} \to \mathcal{Z}$ to map each observation–auxiliary-variable pair $(x_s, y_s)$ to a diffusion *feature latent space* $z_s = g_\phi(x_s, y_s) \in \mathcal{Z}$. This feature latent space is a high-dimensional, near-lossless representation that supports high-fidelity reconstruction through the diffusion decoder $f_\theta : \mathcal{Z} \times \mathcal{Y} \to \mathcal{X}$.

Second, we learn a compact structured embedding $h_\psi : \mathcal{Z} \times \mathcal{Y} \to \mathcal{C}$ such that $c_s = h_\psi(z_s, y_s) \in \mathcal{C}$, designed so that the local geometry of $\mathcal{C}$ aligns with the auxiliary variables and supports smooth, interpretable traversal. While $\mathcal{Z}$ preserves reconstruction fidelity, it remains high-dimensional (e.g., $\dim(\mathcal{Z}) = 32 \times 32 \times 4$ when using cLDM on an image frame with $\dim(\mathcal{X}) = 256 \times 256 \times 3$) and its geometry is not explicitly organized for stable temporal or condition-dependent editing; direct manipulation in $\mathcal{Z}$ is therefore computationally expensive and brittle. In contrast, the *structured embedding space* $\mathcal{C}$ is low-dimensional (e.g. $\dim(\mathcal{C}) < 10$ for spline-based interpolation), whose local geometry aligns with auxiliary variables and explicitly designed for smooth and interpretable trajectory manipulation, enabling efficient and stable control operations.

Third, we perform trajectory editing by traversing the embedding sequence $c_{1:S}$ using an editing operator $\hat{c}_{s'} = \mathcal{T}_\eta(c_s; y_s \to y_{s'})$, to interpolate, extrapolate, or transfer between conditions. The edited embeddings are then lifted back to the feature latent space using a local-neighborhood–preserving kNN decoder $\ell : \mathcal{C} \to \mathcal{Z}$, which reconstructs each edited latent as a weighted combination of nearby training latents in $\mathcal{C}$. Finally, we render the edited sequence by decoding $\hat{x}_{s'} = f_\theta(\hat{z}_{s'}, y_{s'})$ with the diffusion model decoder.

This principled separation of editing and rendering enables stable, interpretable control of spatiotemporal sequences

while preserving the high generative fidelity of diffusion models. It supports faithful interpolation and extrapolation in $\mathcal{C}$, including missing-frame imputation and structured condition control, with high-quality synthesis achieved by lifting edited embeddings back to $\mathcal{Z}$ for decoding.

### 3.2. Encoding/Decoding to Diffusion Latent space

We train a conditional latent diffusion model (cLDM) to encode or decode to *feature latent space*, which operates diffusion process in the latent space $\mathcal{Z}$ of a pretrained VAE instead of pixel space $\mathcal{X}$ which is very high-dimensional. The forward diffusion process progressively adds Gaussian noise to the data and a reverse diffusion iteratively denoises it, using a U-Net architecture. (Rombach et al., 2022) demonstrated that this approach achieves state-of-the-art performance in tasks such as text-to-image synthesis while significantly reducing computational overhead. We denote $E : \mathcal{X} \to \mathcal{Z}$ as encoder and $D : \mathcal{Z} \to \mathcal{X}$ as decoder of the pretrained VAE. We then learn the cLDM by minimizing the following objective

$$\mathcal{L}_{\text{cLDM}} := \mathbb{E}_{E(x_s), y_s, \, \epsilon \sim \mathcal{N}(0,1), \, t} \left[ \|\epsilon - \epsilon_\theta(z_{s,t}, t, y_s)\|_2^2 \right].$$

Here, $\epsilon_\theta$ denote the UNet model (Ronneberger et al., 2015), $t$ is a diffusion timestep, $z_{s,t}$ is a noisy version of the clean input $E(x_s)$, and $y_s \in \mathcal{Y}$ denotes conditional label. This reduces compute cost and helps the model focus on semantic features rather than pixel-level noise.

For diffusion inversion, we employ DDIM (Song et al., 2021; Mokady et al., 2023) to invert a real encoded image $E(x_s)$ to a noisy latent code $z_s = \tilde{g}_\phi(E(x_s), y_s)$ with $g_\phi(\cdot, y_s) = \tilde{g}_\phi(\cdot, y_s) \circ E$. When used as the starting point in the sampling process, this latent code enables reconstruction of the original image as $\hat{x}_s = (D \circ \tilde{f}_\theta)(z_s, y_s) = f_\theta(z_s, y_s)$ (see Appendix B.1 for details). DDIM inversion provides a deterministic mechanism to compute inverted latent $z_s = z_{s,T}$, yielding a latent trajectory $\{z_{s,t}\}_{t=0}^{T}$ that maps cyclically to and from the data sample. This cycle-consistency property ensures that latent edits remain faithful when decoded back into data space.

### 3.3. Structured Embedding and Controlled Generation via ConDA

The complete ConDA pipeline, including diffusion model training (or use of a pretrained model), diffusion inversion, contrastive embedding, trajectory traversal, kNN lifting, and diffusion sampling is summarized in Algorithm 1 (*Training*) and Algorithm 2 (*Generation*) in Appendix B.1.

**Structured Embedding of Diffusion Latents.** We introduce a supervised contrastive learning framework ConDA by learning a map $h_\psi$ that organizes high dimensional diffusion latents $\mathcal{Z}$ into a low-dimensional structured space $\mathcal{C}$ that

encodes ordered trajectories. We train $h_\psi$ using a supervised InfoNCE objective, where positives are defined as $(z_s, z_p)$ belonging to the same label or condition $y_s = y_p$, while negatives are drawn from samples with differing labels. The supervised contrastive loss is given by

$$\mathcal{L}_{\text{InfoNCE}} =$$
$$-\mathbb{E}_{(s)} \left[ \frac{1}{|P(s)|} \sum_{p \in P(s)} \log \frac{\exp\left(\text{sim}(c_s, c_p)/\tau\right)}{\sum_{a \neq s} \exp\left(\text{sim}(c_s, c_a)/\tau\right)} \right],$$
(1)

where $c_s = h_\psi(z_s, y_s)$, $P(s)$ denotes the set of positives for anchor $s$ (same label), $\text{sim}(c_s, c_p) = -\|c_s - c_p\|_2^2$ is the similarity measure, and $\tau$ is a temperature parameter. This loss encourages embeddings with the same condition to cluster together while separating those with different conditions, enforcing that $\mathcal{C}$ encodes dynamics in a condition-aware, low-dimensional structure.

**Trajectory Modeling and Controlled Generation.** Within the space $\mathcal{C}$ we introduce approaches that enable nonlinear dynamic modeling and controlled generation. The goal is to identify directions of movement in latent space that correspond to meaningful changes in observed data, either along the trajectory of a sequence or across distinct classes. We define a map $\mathcal{T}_\eta : \mathcal{C} \times \mathcal{Y} \times \mathcal{Y} \to \mathcal{C}$ that shifts a source embedding $(c_s, y_s)$ towards a target embedding $(c_{s'}, y_{s'})$, while leaving unrelated factors invariant. This is obtained by solving

$$\hat{c}_{s'} = \arg\min_{\mathcal{T}} \mathcal{L}\left(\mathcal{T}_\eta(c_s; y_s \to y_{s'}); c_{s'}\right) + \lambda\, \Omega\left(\mathcal{T}_\eta\right), \quad (2)$$

where $\mathcal{L}$ is a task-specific discrepancy loss that ensures that the generated sample under the edited latent aligns with the target representation, while a regularization term $\Omega(\cdot)$, weighted by $\lambda$, enforces edits to be small. The solution, $\hat{c}_{s'}$ thus reflects the minimal, structured change in the latent space $\mathcal{C}$ needed to achieve faithful, controllable generation $\hat{x}_{s'} = f_\theta(l(\hat{c}_{s'}), y_{s'})$ aligned with the target condition.

(1) **Spline Interpolation.** For sequential data, we assume that embeddings $c_{1:S}$ lie on or near a smooth manifold that captures the intrinsic phase of the sequence. To recover this structure, we fit a a $C^2$-continuous parametric curve $\gamma : \mathbb{R} \to \mathcal{C}$, $\gamma(\alpha) \approx c_s$ for $\alpha = \alpha_s$, to the ordered pairs $\{(\alpha_s, c_s)\}_{s=1}^{S}$, where $\alpha_s$ is a monotone ordering parameter (e.g., normalized index or latent phase). Concretely, parametric spline curve $\gamma$ is obtained by minimizing a regularized least-squares objective that balances data fidelity and curvature: $\min_\gamma \sum_{s=1}^{S} \|\gamma(\alpha_s) - c_s\|^2 + \lambda \int \|\gamma''(\alpha)\|^2 \, d\alpha$, subject to appropriate boundary conditions. This smooth parameterization lets us define edits as motion along the curve. Given a point $c_s$ at coordinate $\alpha_s$, the edit operator

moves by a small phase increment $\Delta\alpha$:

$$\mathcal{T}_\alpha(c_s; y_s \to y_{s+\Delta s}) \;=\; \gamma(\alpha_s + \Delta\alpha). \qquad (3)$$

Choosing $\Delta\alpha > 0$ or $\Delta\alpha < 0$ traverses forward or backward along the trajectory, enabling interpolation of missing states and extrapolation to unobserved phases.

(2) **Taylor Extrapolation (TEX).** In this local extrapolation approach, we estimate the direction of change in $\mathcal{C}$ that moves a source embedding forward along the trajectory by a step size $\Delta s$ using a second-order Taylor expansion around a local coordinate $s$:

$$\mathcal{T}(c_s; y_s \to y_{s+\Delta s}) = c(s + \Delta s) \approx c(s) + \dot{c}(s)\,\Delta s$$
$$+ \tfrac{1}{2}\ddot{c}(s)\,(\Delta s)^2, \qquad (4)$$

where $\dot{c}, \ddot{c}$ are the first and second derivatives of the local trajectory estimated using finite differences. The first order TEX-1 uses only the linear term, while the second order TEX-2 adds $\tfrac{1}{2}\ddot{c}(s)\,(\Delta s)^2$, which captures local nonlinearities of the trajectory. This allows us to navigate the latent space along locally directed trajectories, enabling generation of sequences consistent with the estimated dynamics.

(3) **Classification+KDE-Based Class Traversal.** Given sequences of latent embeddings $\{\,c^{(j)}_{1:S}\,\}_{j=1}^N$, we first train an SVM with an RBF kernel on $\{c^{(j)}\}$ to separate classes (e.g., responder vs. non-responder). To identify navigation directions between classes, we estimate class-conditional densities via KDE: $f_k(u) = \frac{1}{N_k}\sum_{i=1}^{N_k} K\!\left((u - c_i^{(k)})/h\right)$, $k \in \{0,1\}$, where $K(\cdot)$ is a Gaussian kernel, $h$ is the bandwidth, and $\{c_i^{(k)}\}$ are samples from class $k$. We then form the density difference $\Delta f(u) = f_1(u) - f_0(u)$, and detect class-specific peaks by $m_{\text{class0}} = \arg\max_u\big(-\Delta f(u)\big)$, $m_{\text{class1}} = \arg\max_u\big(\Delta f(u)\big)$. The transformation function here is defined as a traversal from the source to the target class along the line connecting these peaks:

$$\mathcal{T}_\eta(c_s^{(\text{class0})}; y_s^{\text{class0}} \to y_s^{\text{class1}}) =$$
$$m_{\text{class0}} + \eta\big(m_{\text{class1}} - m_{\text{class0}}\big), \quad \eta \in [0,1]. \qquad (5)$$

This procedure provides interpretability by linking movements in latent space to density maxima of the two classes, enabling smooth and explainable transitions between non-responder and responders.

## 4. Experiments

**Datasets.** We evaluate on five spatiotemporal domains: fluid dynamics, neural calcium imaging, facial expressions, therapeutic neurostimulation, and neural spike dynamics; see Appendix A for acquisition, simulation, and split details.

(1) **Flow past a cylinder.** 2D incompressible Navier–Stokes with cylinder interactions following the benchmark of Schäfer et al. (1996); simulated in FEniCS (Logg et al., 2012). We render total $N{=}246{,}000$ RGB images (velocity magnitude) of 82 different flow trajectories, with 3000 time points as $y{=}\tau \in [0, 1.5]$. To reduce temporal leakage, we use blocked splits (Roberts et al., 2017).

(2) **Two-photon calcium imaging (PFC).** High-resolution recordings from mouse prefrontal cortex during cognitive flexibility tasks (Spellman et al., 2021). We convert videos to $N{=}706{,}452$ RGB images with $y{=}(v_n, v_t)$ (session and within-session frame).

(3) **DISFA facial expressions.** Standard AU dataset (Mavadati et al., 2013). We use $N{=}261{,}576$ frames of 27 subjects with 12D facial action units as $y{=}\text{AU} \in \{0, \dots, 5\}^{12}$.

(4) **TMS-induced electric fields.** Individualized head models and E-field simulations with SimNIBS (Thielscher et al., 2015); $N{=}569{,}520$ images from 121 patients with coil angles as $y{=}\Theta \in [10°, 360°]$. Used for classification and class-transfer analyses.

(5) **Monkey reach neural spiking.** Premotor population activity during delayed reaches (Churchland and Kaufman, 2022; Pei et al., 2021). The number of training trials for dataset (non-image dynamics) is 2008 with trial length of 140 time bins and recordings from 182 neurons. Following Kapoor et al. (2024), spikes are mapped to 16-D S4 latents; we train conditional diffusion directly on these latents with 2D reach velocity as $y$.

**Experimental setup.** Complete hyperparameters and algorithmic steps appear in Appendix C.1 and Appendix B.1. For images, we use the LDM autoencoder (Rombach et al., 2022) to encode $x_s \in \mathbb{R}^{256\times256\times3}$ to $E(x_s) \in \mathbb{R}^{32\times32\times4}$, train a `UNet2DModel` (Hugging Face `diffusers` (Von Platen et al., 2022)) with 1000 diffusion steps, and apply DDIM inversion to obtain feature latents $z_s$. As a solver baseline, we also implement the reversible REX-RK4 method (Blasingame and Liu, 2026) using the noise-prediction parameterization, coupling parameter 0.9999, and 8 steps. We evaluate Rex-RK4 with both cLDM and the pretrained IsoDiff. Contrastive embeddings are learned with CEBRA (Schneider et al., 2023) (`offset10-model-mse`); we use $d{=}8$ for fluid and $d{=}3$ otherwise (Appendix Fig. 10). We lift $c_s \in \mathbb{R}^d$ using kNN decoder and render via diffusion sampling. The spatiotemporal datasets consist of 82 fluid flow videos (3,000 frames each), 60 calcium imaging videos (approximately 4,800–10,000 frames), and 27 facial expression videos (4,844 frames each). For held-out segment prediction, we split video sequence into $M$ contiguous train-test blocks, each containing a held-out segment of length $n$: Fluid: $M = 50$, $n = 12$; $Ca^{2+}$: $M = 100$, $n = 7$; DISFA: $M = 88, n = 11$.

**Tasks and metrics.** Across datasets, we evaluate on the following tasks: (i) interpolation; (ii) $n$-step-ahead pre-

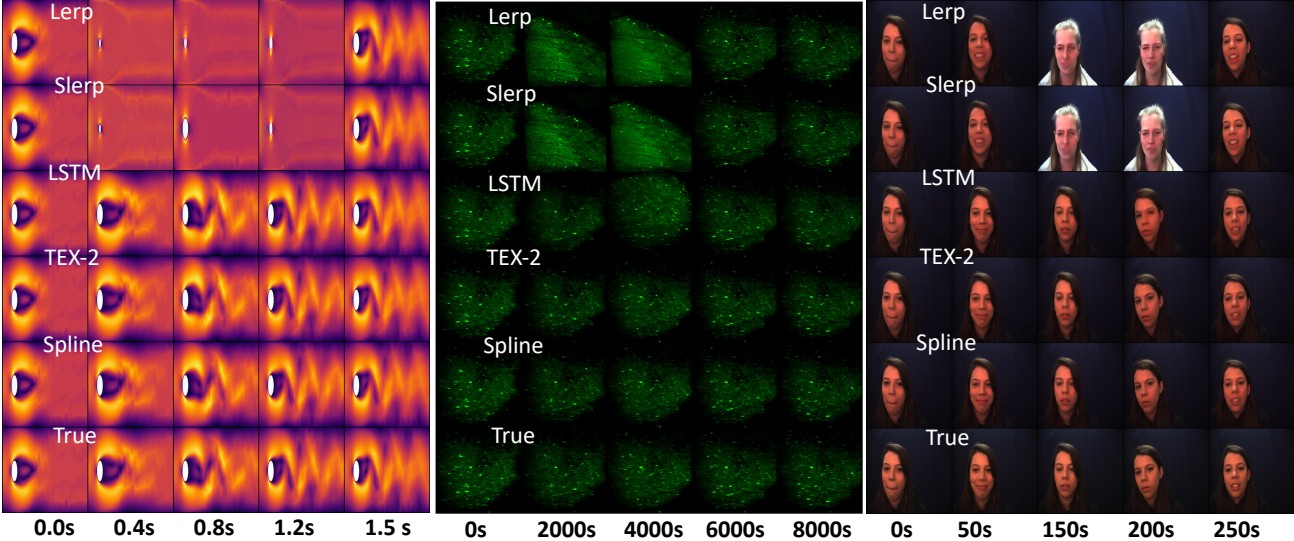

**Figure 2. Visual comparison of generated images from interpolations in $\mathcal{C}$-space.** Linear methods (Lerp, Slerp) show inferior perceptual quality, whereas Spline and TEX-2 yield smoother, more dynamics-consistent reconstructions.

diction; (iii) classification and counterfactual editing in the embedding space; and (iv) interpretability. *Interpolation:* parametric cubic B-splines and finite-difference TEX on latent sequences $z_{1:S}$; baselines are linear (Lerp), spherical (Slerp) (Wang and Golland, 2023; Preechakul et al., 2022), and an LSTM (Hochreiter and Schmidhuber, 1997). *n-step-ahead prediction:* spline extrapolation, TEX-2 and LSTM. *Ablation:* ConDA vs linear (PCA) and variational ($\beta$-VAE, $\beta = 4$) baselines. *Image fidelity:* PSNR and SSIM; *Trajectory error:* RMSE. *Classification:* linear/RBF SVMs with leave-subject-out (TMS) and leave-($Re$)-out (flow) (Osafo Nkansah et al., 2024); we report F1/Acc/AUC. *Class transfer:* KDE interpolation between class-conditional modes in the embedding (details in Appendix C.1). *Interpretability:* on monkey S4 latents, we compare ConDA embeddings to PCA in the same diffusion latents and evaluate prediction on 216 test samples using RMSE, total absolute error (mean/sd), and Procrustes distance.

## 5. Results

We benchmark cLDM against a continuous-label conditioned GAN (Ding et al., 2021) and find that cLDM substantially outperforms the GAN in both image generation and reconstruction (see Appendix C.10 for protocols and full results). In this section, we present both qualitative and quantitative results for our proposed approach ConDA. In addition to full-reference image quality metrics (PSNR, SSIM), we evaluate *trajectory-level* accuracy via RMSE between predicted and ground-truth embeddings.

**Nonlinear Trajectory Modeling: Spline/TEX-2 vs. Baselines.** Our experiments show that the complex nonlinear

dynamics in the high-dimensional diffusion latent space $\mathcal{Z}$ are difficult to model directly: linear baselines underperform, and even nonlinear methods such as LSTM and TEX-1 achieve limited accuracy when operating in $\mathcal{Z}$ (Tab. 1). In contrast, all methods especially Spline and TEX-2 achieve substantially better reconstruction scores when applied in the low-dimensional contrastive space $\mathcal{C}$, validating our approach. Qualitative results (Fig. 2) and generalization tests (Tab. 2, Fig. 11) further confirm that $\mathcal{C}$ captures system dynamics in a more structured, dynamics-aligned geometry, while remaining robust to approximation sources such as kNN lifting, diffusion inversion, and VAE reconstruction (Tab. 8, Appendix C.2). The near-zero RMSE achieved by Spline and TEX-2 in $\mathcal{C}$ highlights its suitability for accurate interpolation, prediction, and stable trajectory editing. We also show in Appendix C.3 that ConDA is solver-agnostic, transferring effectively across diffusion models and inversion solvers (e.g., IsoDiff, Rex-RK4, (Hahm et al., 2024; Blasingame and Liu, 2026)), and that kNN lifting which leverages neighbor-based contrastive structure, consistently outperforms a tuned MLP decoder across datasets (Appendix C.4).

**SVM Classification and KDE-Based Class Traversal.** We evaluated the ConDA embedding space ($\mathcal{C}$) on downstream classification by training linear and RBF-kernel SVMs on two tasks: Fluid (steady vs. unsteady laminar flow) and E-Field (treatment responder vs. non-responder). As shown in Tab. 3, classifiers trained on $\mathcal{C}$-space consistently outperformed those trained directly in the diffusion latent space $\mathcal{Z}$ across Accuracy, F1, and ROC-AUC, independent of the kernel choice, indicating improved separability of task-relevant structure in $\mathcal{C}$. For example, in the E-Field task, an RBF-SVM classifier trained on $\mathcal{C}$ achieved an F1 score of 0.63,

**Table 1. Baseline comparison of interpolations**. We compare spline and TEX-2 (second-order Taylor expansion) against Lerp, Slerp, LSTM, and TEX-1 (first-order Taylor). Linear baselines (Lerp/Slerp) consistently underperform in both $\mathcal{Z}$ and $\mathcal{C}$, indicating inherently nonlinear trajectories. In $\mathcal{Z}$, TEX-2 yields the best fidelity. In $\mathcal{C}$, spline and TEX-2 perform best, consistent with a dynamics-aligned geometry that supports smooth, predictable traversal. Overall, nonlinear methods (LSTM, TEX) improve notably in $\mathcal{C}$ compared to $\mathcal{Z}$.

| Dataset | Method | PSNR ↑ | SSIM ↑ | RMSE ↓ | PSNR ↑ | SSIM ↑ | RMSE ↓ |
|---|---|---|---|---|---|---|---|
| | | DDIM $\mathcal{Z}$-space | | | ConDA $\mathcal{C}$-space | | |
| Fluid | Lerp | $28.16 \pm 0.49$ | $0.56 \pm 0.08$ | 16.12 | $28.27 \pm 0.47$ | $0.59 \pm 0.07$ | 13.75 |
| | Slerp | $28.11 \pm 0.53$ | $0.54 \pm 0.09$ | 17.24 | $28.27 \pm 0.48$ | $0.59 \pm 0.07$ | 13.09 |
| | LSTM | $34.50 \pm 1.81$ | $0.92 \pm 0.09$ | 4.04 | $34.53 \pm 2.05$ | $0.92 \pm 0.08$ | 1.07 |
| | TEX-1 | $29.28 \pm 2.38$ | $0.57 \pm 0.25$ | 14.20 | $32.98 \pm 3.09$ | $0.84 \pm 0.15$ | 3.16 |
| | TEX-2 | $\mathbf{35.29 \pm 0.60}$ | $\mathbf{0.94 \pm 0.01}$ | $\mathbf{0.26}$ | $\mathbf{35.70 \pm 0.36}$ | $\mathbf{0.94 \pm 0.01}$ | 0.02 |
| | Spline | — | — | — | $\mathbf{35.70 \pm 0.36}$ | $\mathbf{0.94 \pm 0.01}$ | $\mathbf{0.00}$ |
| Ca$^{2+}$ | Lerp | $32.36 \pm 1.68$ | $0.79 \pm 0.04$ | 16.60 | $32.92 \pm 2.86$ | $0.82 \pm 0.04$ | 43.87 |
| | Slerp | $32.68 \pm 2.54$ | $0.80 \pm 0.07$ | 17.65 | $32.82 \pm 2.81$ | $0.82 \pm 0.04$ | 43.38 |
| | LSTM | $35.52 \pm 2.70$ | $0.86 \pm 0.07$ | 9.15 | $35.96 \pm 2.91$ | $0.87 \pm 0.07$ | 0.89 |
| | TEX-1 | $30.64 \pm 1.07$ | $0.57 \pm 0.08$ | 22.58 | $36.13 \pm 2.96$ | $0.87 \pm 0.07$ | 0.84 |
| | TEX-2 | $\mathbf{38.00 \pm 0.51}$ | $\mathbf{0.92 \pm 0.01}$ | $\mathbf{0.13}$ | $\mathbf{38.58 \pm 0.59}$ | $\mathbf{0.93 \pm 0.01}$ | $\mathbf{0.00}$ |
| | Spline | — | — | — | $\mathbf{38.58 \pm 0.59}$ | $\mathbf{0.93 \pm 0.01}$ | $\mathbf{0.00}$ |
| DISFA | Lerp | $29.98 \pm 2.70$ | $0.74 \pm 0.08$ | 13.86 | $33.08 \pm 3.25$ | $0.83 \pm 0.09$ | 13.61 |
| | Slerp | $31.13 \pm 2.72$ | $0.76 \pm 0.08$ | 14.97 | $33.13 \pm 3.31$ | $0.83 \pm 0.09$ | 13.68 |
| | LSTM | $38.13 \pm 0.54$ | $\mathbf{0.96 \pm 0.00}$ | 3.64 | $38.74 \pm 0.75$ | $0.96 \pm 0.01$ | 0.34 |
| | TEX-1 | $35.18 \pm 4.06$ | $0.88 \pm 0.14$ | 7.49 | $38.77 \pm 0.70$ | $0.96 \pm 0.00$ | 0.24 |
| | TEX-2 | $\mathbf{38.27 \pm 0.24}$ | $\mathbf{0.96 \pm 0.00}$ | $\mathbf{0.07}$ | $\mathbf{38.99 \pm 0.23}$ | $\mathbf{0.96 \pm 0.00}$ | $\mathbf{0.00}$ |
| | Spline | — | — | — | $\mathbf{38.99 \pm 0.23}$ | $\mathbf{0.96 \pm 0.00}$ | $\mathbf{0.00}$ |

**Table 2. $n$-step-ahead prediction** using nonlinear extrapolation methods (Spline, TEX-2, and LSTM). The results show that generalization remains stable and is not degraded by approximation sources such as k-NN lifting, diffusion inversion, and VAE reconstruction, indicating that these errors are negligible for downstream prediction and do not impede faithful recovery or extrapolation.

| | Fluid | | | Ca$^{2+}$ | | | DISFA | | |
|---|---|---|---|---|---|---|---|---|---|
| Method | PSNR ↑ | SSIM ↑ | RMSE ↓ | PSNR ↑ | SSIM ↑ | RMSE ↓ | PSNR ↑ | SSIM ↑ | RMSE ↓ |
| | ConDA $\mathcal{C}$-space | | | ConDA $\mathcal{C}$-space | | | ConDA $\mathcal{C}$-space | | |
| LSTM | $34.18 \pm 1.97$ | $0.90 \pm 0.11$ | 1.87 | $35.11 \pm 2.57$ | $0.86 \pm 0.06$ | 0.90 | $36.44 \pm 2.80$ | $0.93 \pm 0.06$ | 0.75 |
| TEX-2 | $34.32 \pm 2.31$ | $0.91 \pm 0.10$ | 1.68 | $34.92 \pm 2.53$ | $0.86 \pm 0.06$ | 1.05 | $36.59 \pm 2.71$ | $0.93 \pm 0.07$ | 0.61 |
| Spline | $34.51 \pm 2.13$ | $0.92 \pm 0.07$ | 1.48 | $35.00 \pm 2.57$ | $0.86 \pm 0.06$ | 1.09 | $37.17 \pm 2.17$ | $0.95 \pm 0.04$ | 0.52 |

a marked improvement over the $0.30$ in $\mathcal{Z}$. Leveraging the interpretability afforded by the low dimensionality of $\mathcal{C}$, we visualized the decision-relevant latent dynamics. We used kernel density estimation to model the class-conditional densities and then interpolated along the vector connecting the peaks of these densities. The resulting sequences, depicted in Fig. 3, illustrate the smooth trajectory of change required for a non-responder E-field pattern to morph toward that of a responder. The color-coded changes reveal specific cortical shifts, providing a tangible and interpretable path toward personalizing targeting strategies.

**Table 3. SVM classification performance.** Classification of steady vs. unsteady laminar flow and responder vs. non-responder E-Fields in $\mathcal{Z}$ and $\mathcal{C}$-space.

| | Steady vs. Unsteady | | | Resp. vs. Non-Resp. | | |
|---|---|---|---|---|---|---|
| Method | Acc. ↑ | F1 ↑ | AUC ↑ | Acc. ↑ | F1 ↑ | AUC ↑ |
| | DDIM $\mathcal{Z}$-Space | | | | | |
| SVM-Linear | 0.78 | 0.73 | 0.89 | 0.48 | 0.21 | 0.46 |
| SVM-RBF | 0.68 | 0.52 | 0.59 | 0.57 | 0.30 | 0.58 |
| | ConDA $\mathcal{C}$-Space | | | | | |
| SVM-Linear | 0.83 | 0.81 | 0.82 | 0.62 | 0.62 | 0.68 |
| SVM-RBF | $\mathbf{0.87}$ | $\mathbf{0.85}$ | $\mathbf{0.95}$ | $\mathbf{0.66}$ | $\mathbf{0.63}$ | 0.67 |

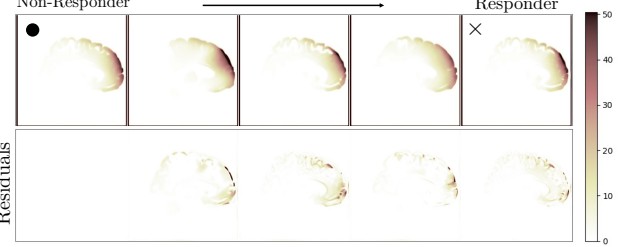

**Figure 3. KDE-based class interpolation.** Trajectory morphing a non-responder E-Field (●) toward a responder (×) by interpolating between class-conditional KDE peaks; colors indicates per-pixel E-field changes from the source, highlighting cortical shifts needed for responder-like patterns and suggesting a path to personalized targeting.

**Ablation: ConDA vs. linear and variational baselines.** We compared ConDA against two baseline representation learning techniques: PCA, a linear method, and a $\beta$-VAE, a variational method on test-segments prediction. To ensure a fair comparison, all models used identical train/validation splits and were configured with the same latent dimension. Each representation was evaluated with its corresponding decoder. As summarized in Tab. 4, ConDA consistently achieved superior performance across all datasets, as mea-

sured by PSNR, SSIM, and RMSE. Qualitative comparisons on DISFA (Fig. 4) further illustrate ConDA's ability to recover fine facial motions at unseen frames with higher fidelity. These results indicate that the baseline models fail to capture the dynamics-relevant variance as effectively as ConDA.

**Table 4. ConDA vs linear and variational baselines.** Performance metrics for test sequence prediction comparing ConDA to PCA and $\beta$-VAE. ConDA achieves improved performance indicating closer alignment with the underlying dynamical manifold.

| Dataset | Method | PSNR ↑ | SSIM ↑ | RMSE ↓ |
|---------|--------|--------|--------|--------|
| Fluid | PCA | $31.82 \pm 2.30$ | $0.87 \pm 0.11$ | $10.43$ |
| | $\beta$-VAE | $32.20 \pm 2.55$ | $0.85 \pm 0.14$ | **7.47** |
| | ConDA | **$33.90 \pm 2.52$** | **$0.90 \pm 0.11$** | $7.60$ |
| $Ca^{2+}$ | PCA | $34.73 \pm 2.29$ | $0.86 \pm 0.05$ | $12.40$ |
| | $\beta$-VAE | $34.62 \pm 2.40$ | $0.85 \pm 0.06$ | $11.58$ |
| | ConDA | **$35.69 \pm 2.54$** | **$0.87 \pm 0.05$** | **11.02** |
| DISFA | PCA | $34.82 \pm 1.03$ | $0.92 \pm 0.02$ | $7.60$ |
| | $\beta$-VAE | $35.76 \pm 1.23$ | $0.94 \pm 0.02$ | $6.59$ |
| | ConDA | **$36.26 \pm 1.03$** | $0.94 \pm 0.01$ | **5.50** |

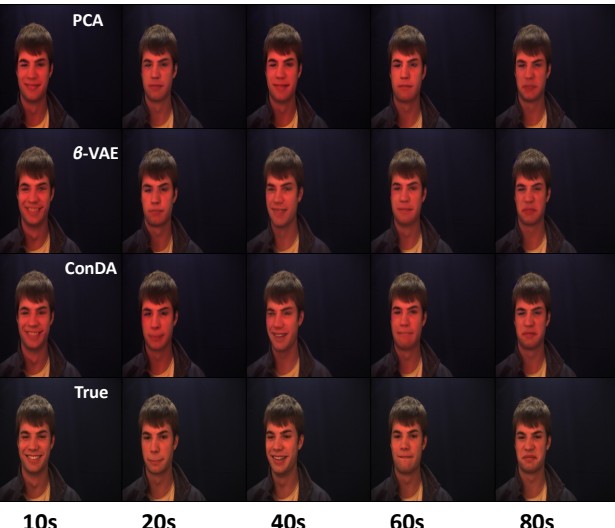

**Figure 4. Comparison of facial expressions at test frames.** ConDA captures facial expressions (e.g., lip-corner pull/stretch) at test frames with higher fidelity than PCA and $\beta$-VAE.

**Interpretability of ConDA-space $\mathcal{C}$.** We assess interpretability on neural spiking data from a monkey performing a delayed center–out reach task (Kapoor et al., 2024). Following the LDNS pipeline, 182-channel spikes are encoded into 16-dimensional, time-aligned latents using an S4-based autoencoder. Conditional diffusion models are trained on these latents, conditioned on 2D hand velocity, with DDIM used for inversion. We visually compare predicted test embeddings in 2D PCA and $\mathcal{C}$ spaces, evaluate alignment with held-out spiking data via RMSE, total absolute error (mean/std), and geometric measures (Procrustes distance). As shown in Fig. 5, $\mathcal{C}$ embeddings exhibit clearer

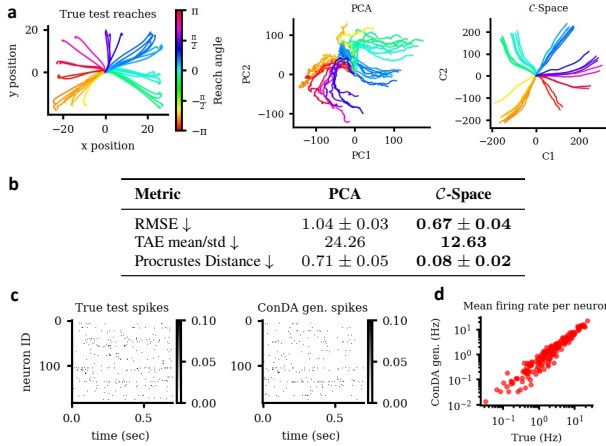

**b**

| Metric | PCA | $\mathcal{C}$-Space |
|--------|-----|---------------------|
| RMSE ↓ | $1.04 \pm 0.03$ | **$0.67 \pm 0.04$** |
| TAE mean/std ↓ | $24.26$ | **12.63** |
| Procrustes Distance ↓ | $0.71 \pm 0.05$ | **$0.08 \pm 0.02$** |

**Figure 5. Interpretability of $\mathcal{C}$ on monkey reach spiking data.** Using LDNS (Kapoor et al., 2024) with structured state-space (S4) layers, we map neural spikes to time-aligned latents and train diffusion models conditioned on reach velocity. **a)** True test reaches, along with predicted test embeddings of neural spikes in 2D PCA and $\mathcal{C}$ spaces. **b)** Performance metrics for PCA and ConDA on test data prediction. **c)** A true and ConDA generated sample of neural activity. **d)** Mean firing rate per neuron, averaged across all trials and time, showing a close match between ConDA predicted firing rates and the real monkey data.

organization by reach direction and stronger alignment with true velocity trajectories than PCA. Quantitative metrics consistently favor $\mathcal{C}$, indicating that contrastive structuring better preserves task-relevant kinematics and improves interpretability relative to variance-only PCA embeddings. Predicted mean firing rates closely match held-out data (Pearson $r = 0.95$; Fig. 5d).

The near-orthogonality of the ConDA latent subspace directions associated with temporal variations and those influenced by confounding factors (e.g., Reynolds number in the fluid dataset) is provided as evidence of successful disentanglement in Appendix C.9.

# 6. Conclusion and Limitations

We introduced ConDA, a contrastive alignment layer that organizes diffusion latents into compact, dynamics-aware embeddings, enabling smooth nonlinear traversals that outperform raw latent and linear baselines across physical and biological domains. ConDA is solver-agnostic and operates consistently across different inversion solvers (e.g., DDIM, Rex-RK4 (Blasingame and Liu, 2026)) as well as across different diffusion models (e.g., IsoDiff (Hahm et al., 2024) latents), see Appendix C. The method has three **limitations**: (1) the embedding is lossy and many-to-one, which we mitigate using a neighborhood-preserving kNN lifting (with learned decoders as future work); (2) the embedding emphasizes local rather than global geometry, which may distort long-range traversals, though locally fitted interpolators and models such as IsoDiff can help; and (3) the approach re-

lies on DDIM inversion, adding computational cost that can be reduced through caching, multi-scale pipelines, or faster solvers such as Rex-RK4 or consistency models. Despite these trade-offs, approximation errors remain small in practice, and ConDA provides a practical path toward controllable and interpretable modeling of complex dynamical systems.

## Impact Statement

This paper proposes a method to enhance the latent space of diffusion models to support interpretable and controllable editing by aligning latent geometry with underlying dynamical factors. Our work shares the same ethical issues as generative models more broadly, including risks related to fake data generation, and malicious manipulation of images, or spatiotemporal signals. While ConDA does not introduce new generative capabilities beyond existing diffusion models, a more structured and disentangled latent representation may also lower the barrier for certain forms of misuse. At the same time, ConDA is designed primarily for scientific and biomedical applications, such as modeling physical systems, neural activity, and therapeutic neurostimulation, where interpretability and controlled counterfactual reasoning are critical. Ethical concerns related to dataset bias, representational fairness, and downstream discrimination remain unchanged by our approach; ConDA neither mitigates nor exacerbates these issues directly. We believe that responsible deployment, transparency about limitations, and domain-expert oversight are essential when applying such methods in high-stakes settings. A collective effort within the research community and society will be important to ensure that generative models and their extensions remain beneficial and are used in socially responsible ways.

## Acknowledgments and Disclosure of Funding

LG is supported by NIH R01MH131534, NIH R01MH118388, the New Venture Fund 202423, the Whitehall Foundation grant (WF 2021-08-089), the Cornell Center for Pandemic Prevention Research seed grant, and an A2 Collective pilot grant (PennAITech, NIA). CL is supported by grants from the National Institute of Mental Health, Wellcome Leap, and the Hope for Depression Research Foundation. We acknowledge the Weill Cornell Medicine Scientific Computing Unit for providing computing resources that supported this work.

## Datasets and Code Access Information

Our source code is available at https://github.com/Grosenick-Lab-Cornell/ConDA. We release the Flow Past a Cylinder benchmark dataset on Hugging Face at https://huggingface.co/datasets/ruchi-sandilya/Flow-Past-Cylinder. Full details of the FEM simulation procedure are provided in Section A.1, with simulation code available at https://github.com/ruchi-sandilya/Flow_past_a_cylinder_benchmark. The two-photon calcium imaging data are available upon request from the authors of the original study (Spellman et al., 2021). The DISFA dataset is available upon request through the official dataset website https://www.mohammadmahoor.com/pages/databases/disfa/. The monkey reaching dataset, MC_Maze, is publicly available through the DANDI Archive under Dandiset ID 000128 and a CC-BY-4.0 license. It contains sorted single-unit spike times and synchronized behavioral measurements recorded from primary motor and dorsal premotor cortex during a delayed reaching task. Details for generating the E-field data are provided in A.2. However, we cannot provide the HAM-D scores or real patient neuroimaging data because these data are part of an ongoing clinical trial and are HIPAA-protected.

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

# A. Data Acquisition

## A.1. Simulation of Flow Past a Circular Cylinder

The flow past a cylinder is a fundamental fluid dynamics problem with numerous practical applications in engineering, science, and industry. As the Reynolds number $Re$ increases, the interesting nonlinear phenomenon of Karman vortex shedding occurs and the flow becomes time-periodic with vortices shedding behind the cylinder. For low Reynolds numbers, the flow remains stationary.

For flow around a circular cylinder, a two-dimensional model is sufficient to capture the essential flow features. The underlying flow geometry and boundary conditions are illustrated in Figure 6. Assuming a fluid density of $\rho = 1.0$, the fluid dynamics are governed by the non-stationary Navier-Stokes equations:

$$\mathbf{u}_t - \nu \Delta \mathbf{u} + \mathbf{u} \cdot \nabla \mathbf{u} + \nabla p = 0, \quad \nabla \cdot \mathbf{u} = 0$$

where $\mathbf{u}$ represents the velocity and $p$ the pressure. Here the kinematic viscosity is set to $\nu = 0.001$. No-slip boundary conditions are applied to the lower and upper walls, as well as the boundary of the cylinder. On the left edge, a parabolic inflow profile is prescribed:

$$\mathbf{u}(0, y) = \left( \frac{4U y (0.41 - y)}{0.41^2}, 0 \right)$$

with a maximum velocity $U = \frac{3\nu Re}{4r}$, where $Re$ and $r$ denote the Reynolds number and the radius of the cylinder, respectively. On the right edge, do-nothing boundary conditions define the outflow:

$$\nu \frac{\partial \mathbf{u}}{\partial n} - p\mathbf{n} = 0$$

with $\mathbf{n}$ representing the outer normal vector.

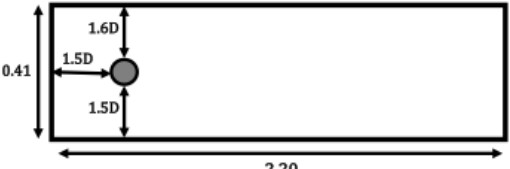 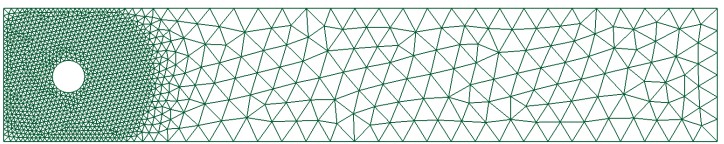

**Figure 6.** (Left) The geometry utilized in fluid simulations, envisioned as a pipe without a cylindrical structure with diameter $D = 2r$. (Right) An adaptively refined mesh with 1446 nodes and 2897 element for fluid flow simulations.

We use FEniCS (Logg et al., 2012), a Finite Element Method (FEM) library, to solve the governing Navier-Stokes equations on an adaptively refined mesh for spatial discretization as shown in Figure 6. Our simulations generate a dataset with four groups exhibiting different flow behaviors based on their input parameters: the first group has Reynolds numbers ($Re$) ranging from 20 to 40 with a cylinder radius ($r$) of 0.05 meters; the second group has $Re$ ranging from 100 to 120 with $r = 0.05$ meters; the third group has $r$ ranging from 0.01 to 0.05 meters with $Re = 20$; and the fourth group has $r$ ranging from 0.05 to 0.1 meters with $Re = 120$. The simulations cover a time range from 0 to 1.5 seconds with a time step size of 0.005 seconds, capturing the persistent behavior of the system, whether characterized by stationary or periodic states.

**Transforming Fluid Flow Simulations into Image Space.** After obtaining the FEM solution for the flow past a circular cylinder, we use the plot function from DOLFIN, which is based on Python libraries such as Matplotlib, to visualize the velocity magnitude at a given time step. This function converts floating-point simulation data into an RGBA image representation. Additional customization and saving of the plots as image files are handled using Matplotlib, resulting in a dataset of $256 \times 256$ pixel images.

## A.2. Simulation of TMS-induced Electric Field

Since the early stages of Transcranial Magnetic Stimulation (TMS), field calculations have been employed for designing coils and, more recently, for assessing the spatial stimulation pattern induced by TMS stimulation in the brain. These calculations are vital for enhancing the precision, effectiveness, and safety of TMS interventions, and they serve a central role

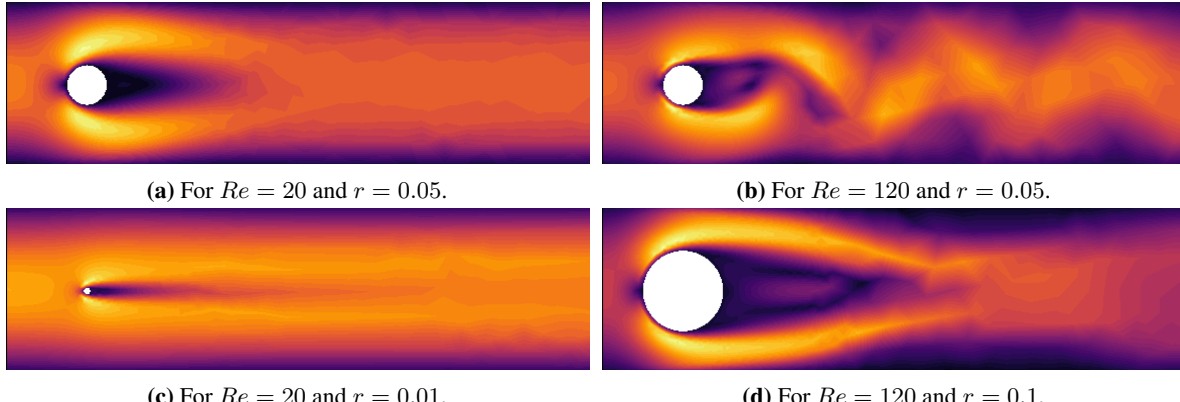

**(a)** For $Re = 20$ and $r = 0.05$.  **(b)** For $Re = 120$ and $r = 0.05$.

**(c)** For $Re = 20$ and $r = 0.01$.  **(d)** For $Re = 120$ and $r = 0.1$.

**Figure 7.** Diverse flow characteristics observed across various flow groups at time 1.5 secs. Note the aspect ratio was changed to yield square images for the analyses in the main text (allowing comparisons with CcGAN, which requires square images).

in advancing our understanding of the neural mechanisms behind TMS efficacy and in developing personalized treatment strategies.

We use a volume conductor model to simulate the electric field distribution in a patient's brain during TMS. Such a model factors in tissue conductivity, individualized head anatomy, and TMS coil parameters, optimizing predictions based on variations in skull thickness and tissue boundaries. Rooted in Maxwell's equations, the model accounts for the coil's size, shape, and orientation, as well as stimulation parameters, influencing the strength and characteristics of the induced electric field. Most of the tools that are available for realistic field calculations rely on methods such as FEM and head models that accurately capture the important anatomical features. We use the SimNIBS library (Thielscher et al., 2015) for head model reconstruction and for personalized E-Field simulations. We first create a subject-specific tetrahedral head mesh from T1-weighted and T2-weighted magnetic resonance (MR) scans using a bash script "charm" (Puonti et al., 2020). The generation of the head model is the most time-consuming step and takes $\approx 3$ hrs. The final meshes contain around $612,278$ nodes and $3,610,350$ tetrahedra (see Fig. 8), divided into different tissue classes. After the generation of a volume head model, FEM solvers are used to calculate the cortical distribution of electric fields in response to different TMS coil positions, intensities, and orientations.

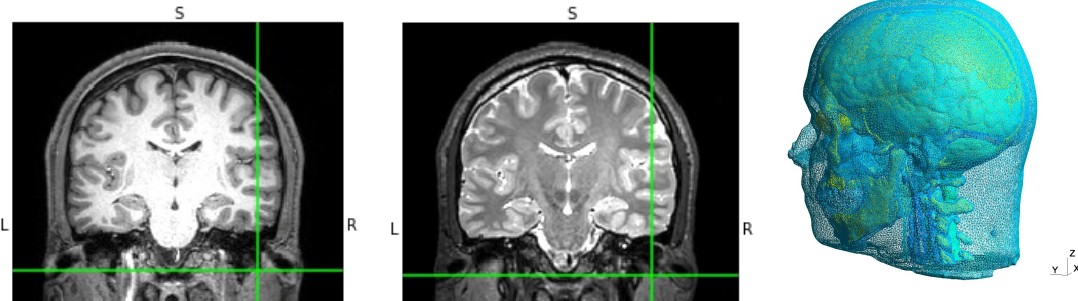

**Figure 8.** Illustration of T1w (left) and T2w (center) images of a representative subject from SimNIBS used for head model reconstruction. The right panel shows the detailed, subject-specific head mesh used for personalized E-Field simulations.

TMS employs time-varying magnetic fields (B-fields) to induce electric fields (E-fields) according to Maxwell-Faraday law (Thielscher et al., 2011; Opitz et al., 2011; Daneshzand et al., 2021) $\nabla \times \mathbf{E} = \frac{\partial \mathbf{B}}{\partial t}$. The total E-Field has the general form $\mathbf{E} = \frac{\partial \mathbf{A}}{\partial t} - \nabla \varphi$, where the first term $\frac{\partial \mathbf{A}}{\partial t}$ involving the magnetic vector potential $\mathbf{A}$ corresponds to the primary field (excitation) induced by the current in the coil, whereas the term $\nabla \varphi$ involving the scalar potential $\varphi$ is called the secondary field. The primary and secondary fields are coupled through the condition of volumetric quasi-neutrality $\nabla \cdot \mathbf{J} = \nabla \cdot (\sigma \mathbf{E})$ where $\mathbf{J}$ denotes the current density and $\sigma$ denotes the electric conductivity of the tissue. The secondary field is generated by charge accumulation at the conductivity boundaries to render the normal component of the current continuous (Miranda

et al., 2007). The FEM solver implements the Galerkin method based on tetrahedral first order elements to determine $\varphi$ at the nodes. The electric fields and current densities are determined in each mesh element based on the above equations. One simulation takes around 6 minutes and uses a maximum of $\sim$4 GB memory.

Our simulations aim to generate TMS induced E-field datasets of depressed patients with the coil centered at their dorsolateral prefrontal cortex (DLPFC) coordinates with different coil angles. The goal is to study the performance of our method in generalizing the distributions of E-Field over different coil angles and different brain regions in real time. We consider simulating E-Fields mapped to NIfTI volume slices which accounts for each patient's unique functional neuroanatomy and cortical folding patterns. The coil is centered at the DLPFC coordinate of an individual pointing posteriorly towards different angles $y = \Theta \in (0°, 360°)$ with angle resolution $\Delta\Theta = 10°$ within a circle formed by considering gray matter vertices within a distance of 20mm and clustering the vertices into $360°/\Delta\Theta$ clusters using k-means.

For model training, we consider T1w and T2w images of $n = 121$ treatment resistant depressed patients undergoing accelerated trials at Weill Cornell Medicine, Cornell University for head model reconstruction. We use their DLPFC coordinates as the coil target and consider coil orientations determined by the $360°/\Delta\Theta$ different angles. The coil model used was Magventure Cool-B65 and therefore we set stimulation intensity to $80A/\mu s$ (Drakaki et al., 2022) consistent with TMS treatment levels. The coil-to-cortex distance was set to $4$ mm.

**Transforming E-Field Simulations into Image Space.** The outcomes of E-Field simulations are saved in both Gmsh and NIfTI formats, with the latter mapping the E-Field values onto subject-specific volumes. To process this data, we utilize the NiBabel Python package, which allows us to access the E-Field values mapped onto NIfTI volumes as NumPy arrays. These arrays match the orientation and dimensions of 3D brain imaging data. For visualization, we generate 2D images by slicing through the 3D imaging data volume, with each slice representing a distinct cross-sectional view of the subject at a precise location. Using Matplotlib, we then save these visualized E-Field mappings as RGBA image files. Through this method, we effectively convert E-Field simulation outputs into image data ($256 \times 256$ pixels).

**Identifying as Responders or Non-Responders.** In our nonlinear classification analysis, to classify patients as responders or non-responders we analyzed the HAMD-17 scores of $n = 110$ subjects for which we had pre- and post-treatment HAMD-17 scores and who received TMS treatment targeted to DLPFC, calculating the percentage change from baseline using the formula:

$$\frac{\text{HAMD-17}_{\text{after}} - \text{HAMD-17}_{\text{before}}}{\text{HAMD-17}_{\text{before}}} \times 100,$$

where $\text{HAMD-17}_{\text{before}}$ and $\text{HAMD-17}_{\text{after}}$ represent the scores before and after treatment, respectively. Patients who exhibited a decrease of 50% or more in their HAMD-17 scores were classified as responders, while those with less than a 50% reduction were classified as non-responders (consistent with prior work (Cole et al., 2020)), see Figure 9.

### A.3. 2-Photon Calcium Imaging Datasets of Mouse PFC

Tracking the activity of specific neuronal populations in the prefrontal cortex (PFC) involved in cognitive flexibility provides important insights into the neural mechanisms underlying such behavior. In this work, we use a comprehensive dataset from (Spellman et al., 2021), which includes high-resolution calcium imaging data capturing PFC neuronal dynamics in mice as they perform behavioral tasks that require flexible cognitive control.

Two-photon laser scanning microscopy was employed to image pyramidal neurons expressing the genetically encoded calcium indicator GCaMP6f. The imaging field of view was designed to preserve cortical laminar structure and included both the prelimbic and infralimbic regions of the PFC. Calcium signals were acquired at a resolution of $256 \times 130$ pixels, spanning a $1500, \mu\text{m} \times 760, \mu\text{m}$ area, corresponding to a spatial resolution of $5.85, \mu\text{m}$ per pixel. Each scan lasted 346 milliseconds, yielding a frame rate of 2.89 Hz. The dataset comprises video recordings from 21 mice, with approximately three imaging sessions per animal and a total of 1,462 behavioral trials. To enable cross-session analysis of the same neuronal populations, non-rigid co-registration was performed using Python package CellReg.

The behavioral paradigm involved a structured sequence of task transitions designed to assess different aspects of cognitive flexibility. These included: simple discrimination (SD), where mice discriminated between two stimuli within a single sensory modality; compound discrimination (CD), where an irrelevant stimulus from a different modality was added; intradimensional shift (IDS), which introduced new exemplars within the same modality; reversal (Rev), where the stimulus-response mapping was switched; extradimensional shift (EDS), where the relevant modality was changed for the first time; a second IDS (IDS2), introducing new exemplars within the newly relevant modality; and serial extradimensional

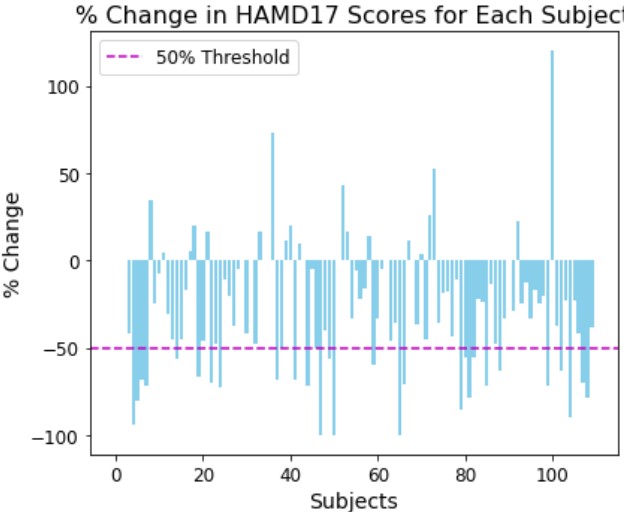

**Figure 9. Percent change in HAMD-17 scores.** Dashed line represents the threshold to classify a patient as a treatment responder, defined as a percent change of $\leq 50\%$. Based on this threshold, 30 patients are identified as treatment responders, while 80 patients are identified as non-responders

shift (SEDS), where the task rule switched automatically upon reaching performance criterion. Throughout this paradigm, task-related neural activity was recorded using two-photon calcium imaging through a coronally implanted microprism, allowing simultaneous visualization of prelimbic and infralimbic areas while preserving laminar architecture. Further methodological details are provided in (Spellman et al., 2021).

To train the conditional Latent Diffusion Model (cLDM), the calcium imaging data were transformed into RGB images of resolution $256 \times 256$ pixels. Each image was conditioned on a frame identifier $v_n \in 0, 1, \ldots, 59$ and a trial time point $v_t \in 0, 1, \ldots, 19999$, forming the conditioning variable $y = (v_n, v_t)$. In total, 706,452 images were generated and used for training and validating the cLDM model.

### A.4. DISFA Dataset

As a facial expression dynamics dataset, we use DISFA dataset (Mavadati et al., 2013), a widely used benchmark in affective computing, facial action unit (AU) recognition, and temporal emotion analysis. It consists of high-resolution spontaneous facial expression videos recorded from 27 subjects while they watched video stimuli designed to elicit natural emotional responses. Each video sequence is annotated frame-by-frame with Facial Action Units (AUs) following the Facial Action Coding System (FACS).DISFA contains approximately 4,844 frames (recorded at 20 frames per second) for each subject, each with intensity scores for 12 AUs: AU1 (inner brow raise), AU2 (outer brow raise), AU4 (brow lowerer), AU5 (upper lid raise), AU6 (cheek raise), AU9 (nose wrinkler), AU12 (lip corner puller), AU15 (lip corner depressor), AU17 (chin raiser), AU20 (lip stretcher), AU25 (lips part), and AU26 (jaw drop). Intensities are discretized on a 6-point ordinal scale from $\{0, 1, 2, 3, 4, 5\}$ where 0 indicates absence and 5 denotes maximum intensity. For our experiments, we transform video frames into 261,576 RGB images (from both left and right video camera) and pair each frame with its AU intensity vector, giving condition labels $y = \text{AU} \in \{0, 1, 2, 3, 4, 5\}^{12}$.

# B. Method

---

**Algorithm 1 Diffusion Model Training + Diffusion Inversion + ConDA Training**

---

**Inputs:** Training data $\{(x_s, y_s)\}$, noise schedule $\alpha_t$, pretrained VAE encoder $E$, distance metric $d(\cdot, \cdot)$, temperature $\tau$, positive window $\Delta$

**Outputs:** Trained diffusion model $\epsilon_\theta$, diffusion latents $z_T$, trained contrastive encoder $h_\psi$

---

**Stage A: Diffusion Model Training ($\epsilon_\theta$) (e.g., cLDM, IsoDiff, AE-s4; model may be pre-trained)**

---

Initialize $z_0 = E(x_s)$
**repeat**
    Sample $z_0 \sim p(E(x) \mid y)$
    Sample $t \sim \text{Uniform}\{1, \ldots, T\}$
    Sample $\epsilon \sim \mathcal{N}(0, I)$
    Compute $z_t = \sqrt{\bar{\alpha}_t}\, z_0 + \sqrt{1 - \bar{\alpha}_t}\, \epsilon$
    Take gradient step on $\|\epsilon - \epsilon_\theta(z_t, t, y_s)\|^2$
**until** convergence
Fix UNet $\epsilon_\theta$

---

**Stage B: Diffusion Inversion ($g_\theta$) (e.g., DDIM, Rex-RK4)**

---

Initialize $z_0 = E(x_s)$
**for** $t = 1, \ldots, T$ **do**
    DDIM inversion: $z_t = \dfrac{\sqrt{\alpha_t}\,(z_{t-1} - \sqrt{1 - \alpha_{t-1}}\, \epsilon_\theta(z_{t-1}, t, y_s))}{\sqrt{\alpha_{t-1}}} + \sqrt{1 - \alpha_t}\, \epsilon_\theta(z_{t-1}, t, y_s)$
**end for**
Store inverted latent $z_T = z_s$

---

**Stage C: ConDA (our contribution) ($h_\psi$)**

---

Build training tuples $\{(z_s, y_s)\}$ by sampling frames
**repeat**
    Construct positives: $P(i) = \{\, j : d(y_j, y_i) \leq \Delta \,\}$; negatives: $N(i) = $ all other indices
    Compute embeddings $c_s = h_\psi(z_s, y_s)$ for all items in a minibatch
    Optimize the InfoNCE objective:

$$\mathcal{L}_{\text{InfoNCE}} = -\sum_i \frac{1}{|P(i)|} \sum_{p \in P(i)} \log \frac{\exp(\text{sim}(c_i, c_p)/\tau)}{\sum_{q \in P(i) \cup N(i)} \exp(\text{sim}(c_i, c_q)/\tau)}$$

    where $\text{sim}(u, v) = -\|u - v\|^2$
**until** validation loss saturates
Tune embedding dimension via cross-validation on reconstruction scores
Fix encoder $\hat{h}_\psi = h_\psi$

---

---

**Algorithm 2 Prediction/Traversal + kNN Lifting + Diffusion Sampling (Generation)**

---

**Inputs:** Latent trajectory $(c_1, \ldots, c_s)$, editing operator $\mathcal{T}_\eta$, neighbor count $k^\star$ for kNN decoder, trained DDIM sampler $f_\theta$, pretrained VAE decoder $D$
**Outputs:** Predicted embedding $c_{s'}$, generated sample $\hat{x}_{s'}$

---

**Stage A: Prediction / Traversal**

---

Use traversal approach $\mathcal{T}$ (Spline, TEX-2, etc.)
Estimate local derivatives: $\dot{c}(s) \approx \frac{c_s - c_{s-1}}{\Delta s}, \quad \ddot{c}(s) \approx \frac{c_s - 2c_{s-1} + c_{s-2}}{(\Delta s)^2}$
Predict next latent: $c(s + \Delta s) \approx c(s) + \dot{c}(s)\Delta s + \frac{1}{2}\ddot{c}(s)(\Delta s)^2$
Set $c_{s+1} \leftarrow \mathcal{T}(c_s)$
Repeat recursively for multi-step prediction

---

**Stage B: kNN Lifting ($l$)**

---

Build training pairs $(c_i, z_i)$
For held-out $c$, compute neighbor set: $N_{k^\star}(c) = \{i_1, \ldots, i_{k^\star}\}$
Compute weighted latent estimate: $l(c) = \frac{\sum_{i \in N_{k^\star}(c)} w_i z_i}{\sum_{i \in N_{k^\star}(c)} w_i}, \quad w_i = \frac{1}{d(c, c_i) + \varepsilon}$
Select $k^\star$ via cross-validation over grid $\mathcal{K} = k_1, k_2, \ldots$ using latent-space $R^2$ score
Fix $k^\star$ for inference
Lift predicted embedding: $l(c_{s'}) = z_{s'}$

---

**Stage C: Sampling ($f_\theta$) (e.g., DDIM, Rex-RK4)**

---

Set $z_T = z_{s'}$
**for** $t = T, \ldots, 1$ **do**
    DDIM Sampling: $z_{t-1} = \frac{\sqrt{\alpha_{t-1}}(z_t - \sqrt{1 - \alpha_t}\,\epsilon_\theta(z_t, t, y_{s'}))}{\sqrt{\alpha_t}} + \sqrt{1 - \alpha_{t-1}}\,\epsilon_\theta(z_t, t, y_{s'})$
**end for**
Store $z_0$
Return $\hat{x}_{s'} = D(z_0)$

---

## B.1. Algorithmic Pipeline of the ConDA Framework

In this section, we present **Algorithm 1 (Training)** and **Algorithm 2 (Generation)**, which explicitly formalize the full ConDA pipeline:

1. *cLDM training $\rightarrow$ diffusion inversion $\rightarrow$ ConDA training*, and

2. *Prediction/Traversal $\rightarrow$ k-NN lifting $\rightarrow$ diffusion sampling*.

### B.1.1. RUNNING EXAMPLE OF CONDA WITH DISFA FACIAL EXPRESSION SEQUENCE

In DISFA, each sample is a short video of one subject transitioning from a neutral face to an expressive face and back (e.g., a smile associated with AU12/25 activation; 12 AUs total). Each video consists of a frame sequence $(x_1, \ldots, x_S)$ and corresponding per-frame AU labels $(y_1, \ldots, y_S)$.

We encode every frame $x_s$ into a diffusion latent $z_s$ using DDIM inversion $(g_\theta)$ (Algorithm 1, Stage B) applied to a pretrained latent diffusion model (trained using Algorithm 1, Stage A), treating $\{z_s\}_{s=1}^S$ as the subject's latent trajectory. ConDA then learns a low-dimensional embedding $c_s = h_\psi(z_s, y_s)$ in $\mathcal{C}$-space (Algorithm 1, Stage C) such that geometric distance and direction in $\mathcal{C}$-space align with expression dynamics (e.g., time within the expression cycle or AU intensity). In practice, this

means that moving along a simple one-dimensional curve in $\mathcal{C}$-space corresponds to smoothly increasing or decreasing the strength of a smile.

For generation, we sample a trajectory in $\mathcal{C}$-space (e.g., a linear or spline path from "neutral" to "expressive" and back) (Algorithm 2, Stage A), map it back to diffusion latents via the $k$NN lifting operator ($l$) (Algorithm 2, Stage B), and decode each latent using diffusion sampling ($f_\theta$) to obtain a controllable facial-expression video (Algorithm 2, Stage C). We evaluate predicted trajectories using RMSE in latent space, and evaluate generated frames using PSNR and SSIM.

### B.2. CcGAN Inversion: Encoder Training

For a given generator function $G(\mathbf{z}, y) \sim f_\theta(\mathbf{z}, y)$ of the CcGAN, we train the encoder $E_z(\mathbf{x})$ by minimizing the following loss function:

$$\mathcal{L}_{ez} = \mathbb{E}_{\mathbf{z} \sim p_z, (\mathbf{x}, y) \sim p_{data}} \Big[ \underbrace{\|\mathbf{x} - G(E_z(\mathbf{x}), y)\|_2^2}_{\mathcal{L}_{ez_1}} + \eta \underbrace{\|\mathbf{z} - E_z(G(\mathbf{z}, y))\|_2^2}_{\mathcal{L}_{ez_2}} \Big]. \quad (6)$$

Here, $\mathcal{L}ez_1$ represents the squared reconstruction loss in image space, and $\mathcal{L}ez_2$ is the squared cyclic loss in the latent space. Note that the first objective both minimizes the difference between the input image $\mathbf{x}$ and its reconstruction $G(E_z(\mathbf{x}), y)$ and second objective seeks to minimize cyclic loss between the latent code $\mathbf{z}$ and its round-trip projection through the data space and back to the latent spaces via $E_z(G(\mathbf{z}, y))$. These two objectives are traded off by tunable hyperparameter $\eta \in \mathbb{R}_+$.

## C. Results

### C.1. Experimental Setup

**cLDM Framework.** For cLDM training, we utilize the pre-trained VAE from (Rombach et al., 2022) and `UNet2DModel` from Hugging Face diffusers library (Von Platen et al., 2022). This VAE produces a smaller representation of an input image and then reconstructs the image based on this small latent representation with a high degree of fidelity. It takes in 3-channel images and produces a 4-channel latent representation with a reduction factor of 8 for each spatial dimension. That is, a $3 \times 256 \times 256$ input image will be compressed down to a $4 \times 32 \times 32$ latent. The U-Net model is used to take noisy latent samples, timesteps, and conditional labels as inputs to predict noise in the diffusion process. We utilize a linear multistep noise scheduler from diffusers library (Von Platen et al., 2022) to introduce noise with diffusion steps $T = 1000$ for training the model and for inference. The model is trained for $40,000$ iterations using the Adam optimizer with a learning rate of $10^{-4}$ and a batch size of $64$. For inversion, we apply DDIM inversion with $T = 1000$ steps. As a solver baseline, we also implement the reversible Rex-RK4 method (Blasingame and Liu, 2026) using the noise-prediction parameterization, coupling parameter 0.9999, and 8 steps. We evaluate Rex-RK4 with both cLDM and the pretrained Isometric Diffusion model (Hahm et al., 2024).

**Contrastive learning and controlled generation.** To obtain contrastive embeddings of the sequence of feature latents, we use the CEBRA library (Schneider et al., 2023) with the `offset10-model-mse` architecture. Embedding dimensions ($d$) are tuned (Fig. 10), with $d = 8$ for fluid and $d = 3$ for all other datasets, ensuring $d << \dim(\mathcal{Z})$. Embeddings $c_s \in \mathbb{R}^d$ are mapped back to feature-latent space using `cebra.KNNDecoder` followed by image reconstruction via a diffusion sampler. Trajectory accuracy is measured using RMSE, while full-reference image quality is evaluated using PSNR and SSIM. The number of neighbors in the kNN decoder is tuned to maximize prediction accuracy.

For trajectory modeling, we use parametric B-spline interpolation (`splprep`/`splev` from SciPy, suitable for $d \leq 10$) and TEX (via finite-difference). Experiments are conducted on sequences from: fluid ($Re = 120$, $S = 3000$), $Ca^{2+}$ ($v_n = 9$, $S = 6200$) and DISFA (subject = SN025, $S = 4844$). We compare against linear baselines, linear interpolation (Lerp) and spherical interpolation (Slerp) (Hahm et al., 2024; Preechakul et al., 2022; Wang and Golland, 2023), and a nonlinear baseline, an LSTM (Hochreiter and Schmidhuber, 1997) predicting the next frame from the previous 20 frames. For n-steps prediction ahead, the data are split using $M$ contiguous train/test blocks, each containing $n$-length held-out points: Fluid: $M = 50$, $n = 12$; $Ca^{2+}$ : $M = 100$, $n = 7$; $DISFA$ : $M = 88, n = 11$. We evaluate spline extrapolation, TEX-2, and LSTM forecasting the next n frames from the previous 20 frames. As an ablation, we compared ConDA against linear (PCA) and variational baselines ($\beta$-VAE with $\beta = 4$) on predicting the test data.

For classification, we train linear and RBF SVMs with leave-subject-out CV on the TMS E-Field dataset with 110 subjects(responders: $\leq 50\%$ change in HAMD-17; non-responders: $> 50\%$ change in HAMD-17 shown in Fig. 9) and

leave-($Re$)-out on the fluid dataset (steady: $Re \leq 65$; unsteady laminar: $Re > 65$) (Osafo Nkansah et al., 2024)). Generalization is evaluated on 10 held-out subjects and 8 held-out $Re$ reporting F1, accuracy, and ROC-AUC. For class transfer, we perform KDE-based interpolation (`sklearn.neighbors.KernelDensity`) in five steps between non-responder and responder E-fields at a fixed coil angle $y = 90°$ by locating class-conditional density modes (`skimage.feature.peak-local-max`) and editing embeddings along the vector connecting mode peaks.

**CcGAN Framework.** We also benchmarked our cLDM against CcGAN with soft vicinity (Ding et al., 2021; Miyato and Koyama, 2018) for image generation and reconstruction. After significant testing, we use $40,000$ training iterations using the Adam optimizer with $\beta_1$ set to 0.5, $\beta_2$ set to 0.999, dimension of latent space of GAN set to 256, learning rate set to $10^{-5}$ and batch size set to 64. Once the CcGAN is trained, we train an encoder network for the generator to enable inversion. The encoder network architecture, comprised of eight layers of $Conv2d \rightarrow BatchNorm2d \rightarrow ReLU$, had a bottleneck dimension of 256. The encoder was trained for $40,000$ iterations by minimizing the loss function defined in (6) with $\eta = 0.1$, using Adam optimizer with learning rate $10^{-5}$ and batch size 32.

**Computational Resources.** We conducted all training and evaluation using NVIDIA RTX 6000 GPUs with 24GB of memory. The number of trainable parameters for each model is summarized in Tab. 6, and the runtime for each simulation is reported in Tab. 7. The evaluation runtime was measured using a batch size of 64 samples.

**Table 5.** Model components with their pretraining, training, and fine-tuning stages, along with the corresponding loss functions.

| Component | Used in | Pretrained | Trained | Objective (Loss) |
|---|---|---|---|---|
| VAE | cLDM | Yes | No | KL div.+ Reconst. loss (MSE) |
| U-Net | cLDM | No | Yes | Noise Pred. (MSE) |
| IsoDiff | Diffusion Baseline | Yes | No | Noise Pred. (MSE) + Isometry Loss |
| CEBRA | ConDA | No | Yes | InfoNCE |
| SVM | Classify | No | Yes | Hinge loss |
| $\beta$-VAE | $\mathcal{C}$ Baseline | No | Yes | Reconst. loss (MSE) + $\beta\times$ KL Divergence |
| LSTM | Prediction | No | Yes | Seq. regression (MSE) |
| MLP | $\mathcal{C}$ lifting Baseline | No | Yes | MSE |

**Table 6.** Trainable parameters.

| Model | Model Size |
|---|---|
| CcGAN | 77.48M |
| IsoDiff | 113.67M |
| VAE | 83.65M |
| UNet | 113.68M |
| CEBRA | 6.56M |
| $\beta$-VAE ($\mathcal{Z}$-Space) | 18.89M |
| LSTM ($\mathcal{Z}$-Space) | 13.65M |
| LSTM ($\mathcal{C}$-Space) | 0.20M |
| MLP ($\mathcal{C}$-Space) | 1.12M |

**Table 7.** Methods and runtime.

| Method | Fluid | E-Field | 2P Ca$^{2+}$ | DISFA |
|---|---|---|---|---|
| FEM Simul. | $\sim$70 hrs | $\sim$18 days | – | – |
| CcGAN Train. | $\sim$34 hrs | $\sim$43 hrs | – | – |
| CcGAN Eval. | $\sim$28 sec | $\sim$28 sec | – | – |
| cLDM Train. | $\sim$72 hrs | $\sim$73 hrs | $\sim$57 hrs | $\sim$87 hrs |
| cLDM Eval. | $\sim$5 min | $\sim$5 min | $\sim$5 min | $\sim$5 min |
| DDIM | 2.31 s/samp. | – | 2.31s/samp | 1.29 s/samp. |
| Rex RK4 | 0.10 s/samp. | – | 0.18 s/samp. | 0.09 s/samp. |

## C.2. Diffusion Inversion Solvers: Rex-RK4 vs. DDIM reconstruction

Diffusion-model inversion is inherently numerically unstable, as small perturbations from floating-point noise, discretization, or batching can amplify during the reverse trajectory, a phenomenon well-documented in neural differential equations (Kidger, 2022). Recent work on exact diffusion inversion aims to eliminate this drift under low NFE budgets typical of modern pipelines, spanning ODE-based (Blasingame and Liu, 2026; Wallace et al., 2023; Zhang et al., 2024; Wang et al., 2024) and SDE-based approaches (Blasingame and Liu, 2026; Wu and De la Torre, 2023), though many rely on heuristic, low-order schemes or require caching large parts of the forward trajectory, leaving their mathematical footing limited. A more principled direction develops algebraically reversible integrators, including MALI (Zhuang et al., 2021), reversible adjoint methods (Kidger et al., 2021), higher-order reversible ODE solvers (McCallum and Foster, 2024), and reversible diffusion ODE/SDE solvers (Blasingame and Liu, 2026), which offer improved stability. This context motivates our comparison of DDIM, a standard non-reversible first-order sampler, with Rex-RK4, a modern reversible Runge–Kutta solver, to test whether our method is solver-agnostic and whether reversibility improves inversion stability at the same NFE

budget.

We compare DDIM inversion with the reversible Rex-RK4 integrator to assess reconstruction fidelity and evaluate solver dependence. As shown in Table C.2, both methods closely match the VAE reconstruction. Importantly, Rex-RK4 achieves this performance with only 8 steps, providing a significantly lower NFE cost compared to DDIM (1000 steps) while maintaining near identical reconstruction fidelity.

**Table 8.** Reconstruction performance of DDIM and Rex-RK4 (8 steps), with the VAE reconstruction defining the upper bound. Apparent 0.02–0.07 dB PSNR differences in favor of DDIM or Rex-RK4 arise from numerical noise (floating-point, batching, discretization) rather than genuine improvements. These results confirm that inversion errors are negligible relative to VAE error and do not impact on downstream outcomes.

| Data | Method | PSNR ↑ | SSIM ↑ |
|---|---|---|---|
| Fluid | VAE | 35.87 ± 0.38 | 0.95 ± 0.01 |
| | Rex-RK4 | 35.91 ± 0.38 | 0.95 ± 0.01 |
| | DDIM | 35.94 ± 0.59 | 0.95 ± 0.01 |
| Ca2+ | VAE | 38.61 ± 0.42 | 0.93 ± 0.01 |
| | Rex-RK4 | 38.64 ± 0.41 | 0.93 ± 0.01 |
| | DDIM | 38.63 ± 0.42 | 0.93 ± 0.01 |
| DISFA | VAE | 39.07 ± 0.27 | 0.96 ± 0.00 |
| | Rex-RK4 | 39.13 ± 0.25 | 0.96 ± 0.00 |
| | DDIM | 39.05 ± 0.27 | 0.96 ± 0.00 |

### C.3. Trajectory modeling: cLDM (Rex-RK4) and Isometric diffusion (Rex-RK4)

We compare interpolation approaches using an alternative diffusion inversion method, Rex-RK4, applied to both cLDM and the Isometric Diffusion model. The results in Tables 9 and 10 show that counterfactual behavior remains consistent across DDIM (see Tab. 1) and Rex-RK4 (the latter being much faster with only 8 steps), as well as when using Isometric diffusion (IsoDiff). IsoDiff model learns an isometric latent space directly within the diffusion model; we treat its latents as an additional source and apply ConDA on top of them. IsoDiff alone yields strong interpolation performance in its native latent space. Overall, these results confirm that ConDA does not depend on a particular inversion procedure or diffusion model.

**Table 9.** Baseline comparison of interpolations in $\mathcal{C}$-space using cLDM with Rex-RK4 (8 steps).

| Method | Fluid | | | Ca²⁺ | | | DISFA | | |
|---|---|---|---|---|---|---|---|---|---|
| | PSNR ↑ | SSIM ↑ | RMSE ↓ | PSNR ↑ | SSIM ↑ | RMSE ↓ | PSNR ↑ | SSIM ↑ | RMSE ↓ |
| | **cLDM Rex RK4 $\mathcal{C}$-space** | | | **cLDM Rex RK4 $\mathcal{C}$-space** | | | **cLDM Rex RK4 $\mathcal{C}$-space** | | |
| Lerp | 28.75 ± 0.85 | 0.66 ± 0.08 | 11.90 | 37.13 ± 0.64 | 0.89 ± 0.01 | 34.40 | 34.97 ± 0.70 | 0.87 ± 0.02 | 23.26 |
| Slerp | 28.87 ± 0.92 | 0.67 ± 0.08 | 12.44 | 36.76 ± 0.71 | 0.89 ± 0.01 | 55.31 | 35.04 ± 0.77 | 0.87 ± 0.02 | 24.01 |
| LSTM | 34.13 ± 2.31 | 0.92 ± 0.04 | 0.66 | 38.28 ± 0.52 | 0.92 ± 0.01 | 1.83 | 38.71 ± 0.78 | 0.96 ± 0.01 | 0.51 |
| TEX-1 | 33.35 ± 2.95 | 0.88 ± 0.09 | 3.17 | 38.24 ± 0.55 | 0.92 ± 0.01 | 2.09 | 38.68 ± 1.26 | 0.96 ± 0.01 | 0.73 |
| TEX-2 | **35.57 ± 0.34** | **0.94 ± 0.01** | 0.01 | **38.60 ± 0.58** | **0.93 ± 0.01** | **0.00** | **39.07 ± 0.23** | **0.96 ± 0.00** | **0.00** |
| Spline | **35.57 ± 0.34** | **0.94 ± 0.01** | **0.00** | **38.60 ± 0.58** | **0.93 ± 0.01** | **0.00** | **39.07 ± 0.23** | **0.96 ± 0.00** | **0.00** |

**Table 10.** Baseline comparison of interpolations in $\mathcal{Z}$ and $\mathcal{C}$-space using Isometric diffusion model from (Hahm et al., 2024) with Rex-RK4 (8 steps) on DISFA facial expression datasets.

| Data | Method | PSNR ↑ | SSIM ↑ | RMSE ↓ | PSNR ↑ | SSIM ↑ | RMSE ↓ |
|---|---|---|---|---|---|---|---|
| | | **IsoDiff Rex RK4 $\mathcal{Z}$-space** | | | **IsoDiff Rex RK4 $\mathcal{C}$-space** | | |
| **DISFA** | Lerp | 35.14 ± 0.67 | 0.89 ± 0.02 | 21.49 | 35.11 ± 1.24 | 0.89 ± 0.04 | 31.45 |
| | Slerp | 35.35 ± 0.65 | 0.89 ± 0.02 | 21.51 | 35.46 ± 1.85 | 0.90 ± 0.04 | 35.25 |
| | LSTM | 38.75 ± 1.58 | 0.97 ± 0.02 | 7.53 | 38.30 ± 1.39 | 0.97 ± 0.02 | 1.29 |
| | TEX-1 | 38.70 ± 1.46 | 0.97 ± 0.02 | 9.75 | 38.31 ± 2.27 | 0.97 ± 0.03 | 4.80 |
| | TEX-2 | **42.51 ± 0.02** | **1.00 ± 0.00** | **0.00** | **42.51 ± 0.02** | **1.00 ± 0.00** | 0.01 |
| | Spline | — | — | — | **42.51 ± 0.02** | **1.00 ± 0.00** | **0.00** |

## C.4. ConDA decoders: kNN vs MLP lifting

As a baseline, we compare kNN with an MLP decoder in Tab. 11 that takes a $d$-dimensional contrastive embedding and passes it through two fully connected layers with 256 units and ReLU activations (with optional dropout), followed by a final linear layer that maps to a 4096-dimensional output corresponding to the diffusion latent.

**Table 11.** Comparison of k-NN (which leverages neighbor-based contrastive structure) and MLP lifting from $\mathcal{C}$-space to $\mathcal{Z}$-space. The results show that k-NN consistently outperforms a tuned MLP decoder across all datasets.

| Data | Decoder | PSNR ↑ | SSIM ↑ |
|---|---|---|---|
| Fluid | MLP | $30.10 \pm 1.62$ | $0.78 \pm 0.11$ |
| | k-NN | $\mathbf{33.90 \pm 2.52}$ | $\mathbf{0.90 \pm 0.11}$ |
| $Ca^{2+}$ | MLP | $32.05 \pm 1.02$ | $0.79 \pm 0.05$ |
| | k-NN | $\mathbf{35.01 \pm 2.55}$ | $\mathbf{0.86 \pm 0.06}$ |
| DISFA | MLP | $34.08 \pm 2.76$ | $0.87 \pm 0.06$ |
| | k-NN | $\mathbf{36.77 \pm 2.58}$ | $\mathbf{0.94 \pm 0.06}$ |

## C.5. Effect of Contrastive Embedding Space Dimensionality on Test Sequence Prediction

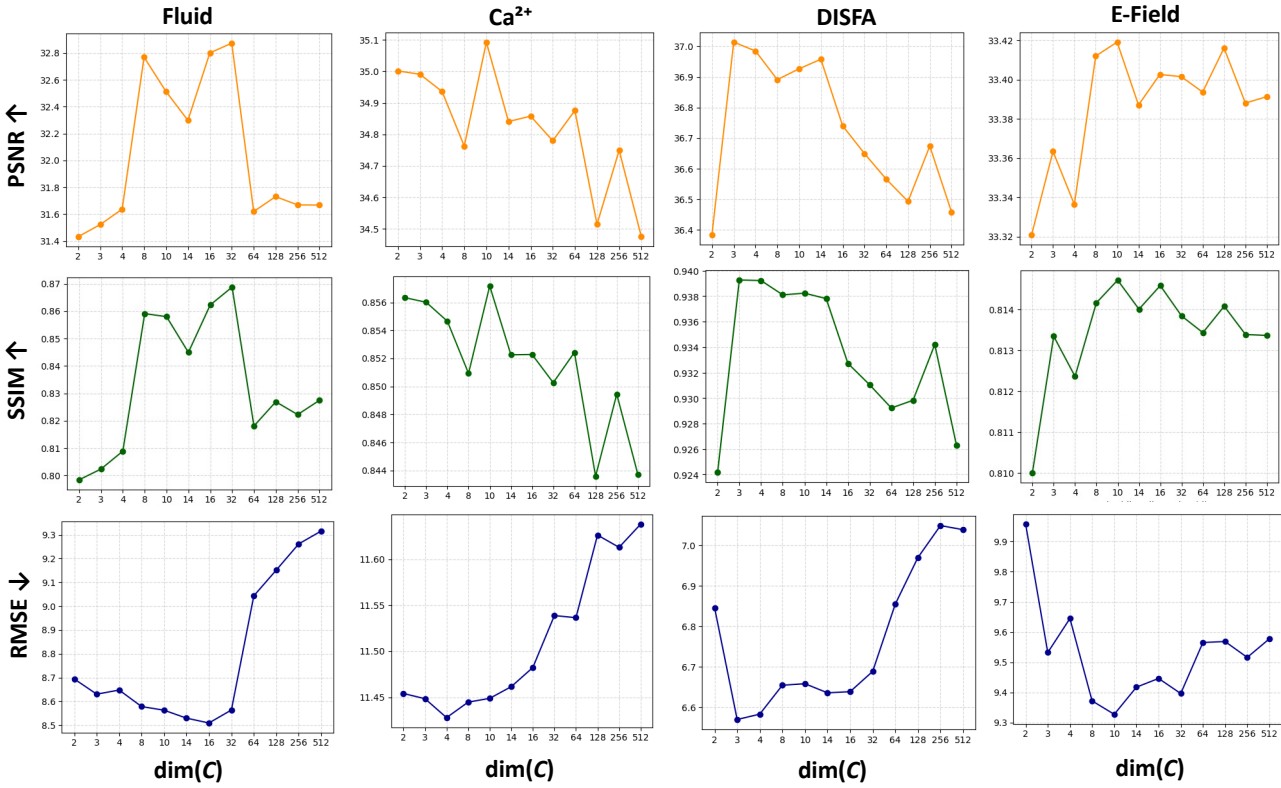

**Figure 10. Prediction performance with embedding dimensionality**. Test-set prediction across $\dim(\mathcal{C}) \in \{2, 3, 4, \cdots, 512\}$ shows negligible degradation for low-dimensional embeddings ($2 < \dim(\mathcal{C}) \leq 10$), indicating minimal information loss.

## C.6. n-Step Ahead Prediction

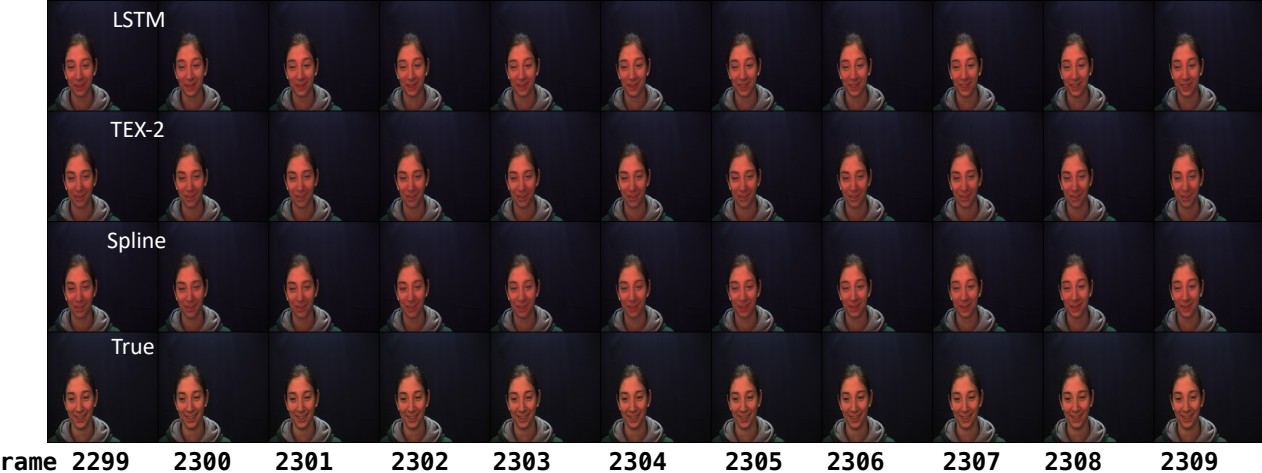

**Figure 11.** $n$-**step forecasting**. Predicted $n = 11$ future frames for a subject in the DISFA facial expression dataset using spline, TEX-2, and LSTM extrapolation. Ground-truth frames are shown in the fourth row. The predicted frames closely match the reference images.
.

## C.7. Condition Interpolation

We report results obtained by varying only the conditioning variables through interpolation while keeping the diffusion latent fixed during sampling, as shown in the Table 12. The results indicate that conditioning-variable interpolation alone does not yield a well-structured control space.

**Table 12.** Condition interpolation vs ConDA-space spline interpolation. Varying only the conditioning variables while keeping the diffusion latent fixed leads to weaker reconstruction quality, indicating that condition interpolation alone does not provide a well-structured control space.

| Method | Fluid | | | Ca$^{2+}$ | | | DISFA | | |
|---|---|---|---|---|---|---|---|---|---|
| | PSNR ↑ | SSIM ↑ | RMSE ↓ | PSNR ↑ | SSIM ↑ | RMSE ↓ | PSNR ↑ | SSIM ↑ | RMSE ↓ |
| Cond. Interp. | $29.44 \pm 1.30$ | $0.74 \pm 0.11$ | 6.58 | $31.53 \pm 1.24$ | $0.74 \pm 0.06$ | 8.29 | $32.29 \pm 2.89$ | $0.79 \pm 0.09$ | 6.90 |
| ConDA (Spline) | $35.70 \pm 0.36$ | $0.94 \pm 0.01$ | 0.00 | $38.58 \pm 0.59$ | $0.93 \pm 0.01$ | 0.00 | $38.99 \pm 0.23$ | $0.96 \pm 0.00$ | 0.00 |

## C.8. Locality Analysis in the ConDA Space

To evaluate the local stability of the learned ConDA space, we perform a locality ablation by perturbing embeddings in $\mathcal{C}$ as $c' = c + \alpha\epsilon$, where $\epsilon \sim \mathcal{N}(0, I)$. We then measure reconstruction quality as a function of the edit distance $d_{\mathcal{C}} = \|c' - c\|$. As shown in Table 13, reconstruction error, measured by RMSE, increases monotonically with distance across all datasets, while PSNR and SSIM degrade smoothly. This indicates that the learned space supports stable local edits and that reconstruction quality deteriorates gradually as perturbations move farther from the data manifold. These results are consistent with the design of the kNN lifting step, which acts as a local, geometry-preserving operator rather than a global inverse mapping from the low-dimensional ConDA space.

**Table 13.** Locality analysis in ConDA space. Stable Diffusion text-to-image generation is included for comparison.

| Distance | Fluid | | | Ca$^{2+}$ | | | DISFA | | |
|---|---|---|---|---|---|---|---|---|---|
| | PSNR ↑ | SSIM ↑ | RMSE ↓ | PSNR ↑ | SSIM ↑ | RMSE ↓ | PSNR ↑ | SSIM ↑ | RMSE ↓ |
| 0.0 | $35.47 \pm 0.53$ | $0.95 \pm 0.01$ | 0.00 | $38.10 \pm 0.49$ | $0.93 \pm 0.01$ | 0.00 | $38.24 \pm 0.37$ | $0.96 \pm 0.00$ | 0.00 |
| 0.6 | $35.46 \pm 0.53$ | $0.95 \pm 0.01$ | 0.68 | $37.74 \pm 0.53$ | $0.92 \pm 0.01$ | 8.63 | $37.75 \pm 1.27$ | $0.96 \pm 0.02$ | 4.93 |
| 1.2 | $35.01 \pm 1.65$ | $0.93 \pm 0.07$ | 4.50 | $37.46 \pm 0.61$ | $0.92 \pm 0.01$ | 12.20 | $37.10 \pm 2.09$ | $0.94 \pm 0.05$ | 7.13 |
| 1.8 | $34.72 \pm 1.89$ | $0.92 \pm 0.08$ | 6.54 | $37.44 \pm 0.54$ | $0.91 \pm 0.01$ | 14.53 | $36.44 \pm 2.77$ | $0.92 \pm 0.07$ | 8.51 |
| 2.4 | $34.19 \pm 2.47$ | $0.90 \pm 0.11$ | 8.09 | $37.35 \pm 0.52$ | $0.91 \pm 0.01$ | 14.51 | $35.79 \pm 3.09$ | $0.90 \pm 0.08$ | 9.41 |
| 3.0 | $33.30 \pm 3.03$ | $0.86 \pm 0.14$ | 9.75 | $36.96 \pm 1.28$ | $0.90 \pm 0.03$ | 15.05 | $35.51 \pm 2.96$ | $0.90 \pm 0.08$ | 10.59 |
| | **Stable Diffusion Text-to-Image** | | | | | | | | |
| 0.0 | $49.88 \pm 3.22$ | $1.00 \pm 0.00$ | 0.00 | | | | | | |
| 0.1 | $49.70 \pm 3.88$ | $0.99 \pm 0.10$ | 2.99 | | | | | | |
| 0.2 | $45.02 \pm 9.85$ | $0.78 \pm 0.41$ | 63.18 | | | | | | |
| 0.3 | $41.12 \pm 11.40$ | $0.60 \pm 0.48$ | 113.64 | | | | | | |
| 0.4 | $35.71 \pm 11.23$ | $0.36 \pm 0.46$ | 184.98 | | | | | | |
| 0.5 | $35.09 \pm 11.05$ | $0.33 \pm 0.46$ | 193.96 | | | | | | |

## C.9. Orthogonality of ConDA Embedding Space

As a key benefit of InfoNCE-based contrastive learning, our ConDA embeddings create a latent space where distinct factors of variation are more orthogonal. To quantify disentanglement, we trained two separate linear regressions to predict the time ($t$) and Reynolds number ($Re$) from the latent embeddings. $\beta_t$ and $\beta_{\mathrm{Re}}$ denote the learned regression coefficient vectors in each respective space. The cosine similarity between these two vectors is reported, with lower values indicating greater orthogonality. This is clearly demonstrated in Tab. 14, which reports the cosine similarity between the latent regression coefficients for time ($\beta_t$) and the confounding Reynolds number ($\beta_{\mathrm{Re}}$) in the fluid dataset. Our ConDA space ($\mathcal{C}$) demonstrates significantly greater orthogonality between these latent directions compared to the raw diffusion latent space ($\mathcal{Z}$), confirming that it better separates temporal variations from confounding physical parameters. This improved separation of factors, temporal variations from physical parameters is crucial for fine-grained control, enabling users to edit the temporal evolution of the system while precisely controlling other physical attributes.

**Table 14. Orthogonality of $\mathcal{C}$.** Cosine similarity between latent regression coefficients for time and Reynolds numbers ($\beta_t$, $\beta_{\mathrm{Re}}$) in fluid dataset. Lower values indicate greater orthogonality.

| Latent Space | Cosine Similarity ($\beta_t, \beta_{Re}$) |
|---|---|
| $\mathcal{Z}$ (dim = 4096) | 0.1014 |
| $\mathcal{C}$ (dim = 3) | 0.0155 |

## C.10. CcGAN vs. cLDM: Generated and Reconstructed Sample Comparison

To assess the quality and diversity of samples generated by CcGAN and cLDM, we performed both qualitative and quantitative analyses on two regression-conditioned datasets: fluid flow conditioned on time ($y = \tau$) and TMS-induced E-field distributions conditioned on coil angles ($y = \Theta$). We provide the visual comparison of generated samples in Fig. 12.

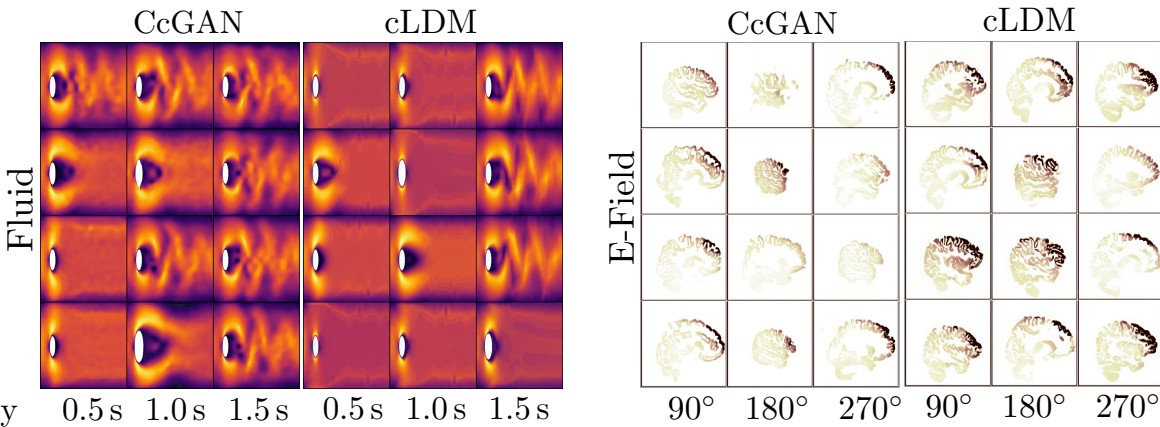

**Figure 12. Visual comparison of randomly sampled images generated by CcGAN and cLDM. (Left)** Generated fluid flow data at different time. **(Right)** Generated E-Field distributions at different TMS coil angles. Rsults indicate cLDM generates high quality samples.

In Fig. 13, the **left** panel presents the standard deviation maps across 60k real and generated images to visualize spatial variability and structural complexity. The cLDM generated samples closely matched the variability of real data, effectively capturing diverse patterns such as varying cylinder radii in fluid flow and heterogeneous cortical foldings in E-Field maps. In contrast, CcGAN exhibited signs of mode collapse, failing to represent key variations, particularly smaller cylinder sizes and patient-specific cortical structures. For quantitative evaluation, we used Fréchet Inception Distance (FID) and Inception Score (IS) to measure image quality and diversity. As summarized in the Table of Fig. 13, cLDM achieved a lower FID and higher IS across both domains, indicating its ability to generate high-quality images with a distribution more consistent with the real data.

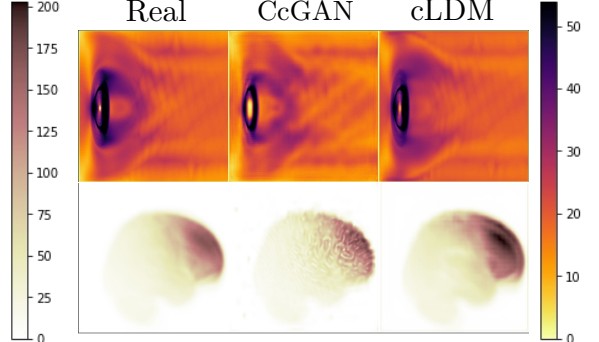

| Dataset | Method | FID ↓ | IS ↑ |
|---------|--------|-------|------|
| Fluid | CcGAN | $0.670 \pm 0.068$ | $1.279 \pm 0.019$ |
|  | cLDM | $\mathbf{0.469} \pm 0.089$ | $\mathbf{1.851} \pm 0.023$ |
| E-Field | CcGAN | $1.920 \pm 0.790$ | $1.435 \pm 0.033$ |
|  | cLDM | $\mathbf{1.246} \pm 0.019$ | $\mathbf{2.083} \pm 0.032$ |

**Figure 13. (Left)** Visual comparison of standard deviation maps computed for each pixel across the sets of 60k real and generated samples with fluid and E-Field data, highlighting the variability and complexity in both sets. Note that CcGAN shows signs of mode collapse (e.g., the cylinder lacks size diversity and E-Field lacks diversity in unique cortical foldings across different patients). Scale bars are normalized velocity (a.u.) and normalized E-Field intensity (a.u.). **(Right)** Quantitative evaluation with lower FID and higher IS for cLDM generated samples indicate better diversity and quality than CcGAN.

Fig. 14 compares reconstructed samples obtained from the CcGAN encoder and DDIM inversion. The top and middle panels display visual reconstructions, while the table in bottom panel summarizes quantitative evaluation using both general-purpose metrics (FID and IS) and domain-specific metrics (Peak Signal-to-Noise Ratio (PSNR) and Structural Similarity Index Measure (SSIM)). These results highlight the improved reconstruction fidelity and better preservation of spatial and semantic structure offered by DDIM-based methods. Overall, cLDM reconstructions yield a more accurate and controllable representation of complex physical systems, supporting their suitability for nonlinear counterfactual generation tasks.

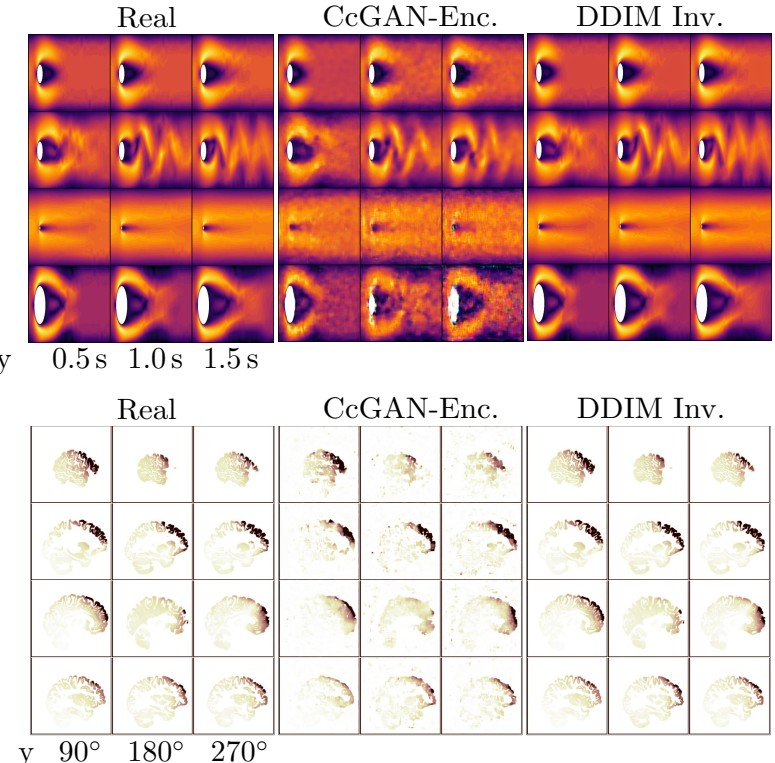

| Dataset | Method | FID ↓ | IS ↑ | PSNR ↑ | SSIM ↑ |
|---------|--------|-------|------|--------|--------|
| Fluid | CcGAN-Enc. | $0.947 \pm 0.137$ | $1.458 \pm 0.079$ | $30.946 \pm 0.387$ | $0.794 \pm 0.047$ |
|  | DDIM Inv. | $\mathbf{0.222 \pm 0.017}$ | $1.458 \pm 0.048$ | $\mathbf{35.791 \pm 0.229}$ | $\mathbf{0.954 \pm 0.004}$ |
| E-Field | CcGAN-Enc. | $1.383 \pm 0.067$ | $1.349 \pm 0.069$ | $34.296 \pm 0.207$ | $0.798 \pm 0.013$ |
|  | DDIM Inv. | $\mathbf{1.274 \pm 0.040}$ | $\mathbf{1.745 \pm 0.076}$ | $\mathbf{36.737 \pm 0.149}$ | $\mathbf{0.955 \pm 0.002}$ |

**Figure 14. Comparison of reconstructed images using CcGAN encoder and DDIM inversion.** Visual assessment of reconstructed samples for two datasets: Different flow behavior conditioned over time **(top)** and TMS induced E-Field distributions conditioned on coil angles **(middle)**. **(Bottom)** Quantitative evaluation for reconstruction performance. Both visual and quantitative results demonstrate superior reconstruction quality using the DDIM-based inversion from the diffusion model.

### C.11. Supplementary Videos

**Supplementary Video 1** presents synthetic video frames generated using the estimated latents from ConDA spline approach to model fluid dynamics at Re=120. The video highlights realistic vortex shedding behavior.

**Supplementary Video 2** presents synthetic video frames generated using the ConDA TEX-2 approach to model 2P $Ca^{2+}$ signal dynamics from the mouse PFC while performing a cognitive task. The video captures both realistic calcium activity dynamics and brain motion.

**Supplementary Video 3** presents synthetic video frames generated using the ConDA prediction of subject's facial expression dynamics at test frames from the DISFA dataset.

