# OpenReview forum: "Contrastive Diffusion Alignment: Learning Structured Latents for Controllable Generation"
_ICML.cc/2026/Conference — ICML 2026 regular_

### Official Review · Reviewer_17dw · 2026-02-24

**Soundness:** 3
**Presentation:** 4
**Significance:** 3
**Originality:** 3
**Overall Recommendation:** 5
**Confidence:** 4

**Summary:**

This paper introduces ConDA (Contrastive Diffusion Alignment), a network architecture designed to address the lack of interpretability in the latent representations of diffusion models. The authors propose a plug-and-play layer that aligns these latents and reconstructs them into meaningful embeddings. Through experiments on five spatiotemporal datasets, ConDA demonstrates consistent improvements over non-aligned baselines, validating the effectiveness of the approach.

**Compliance With Llm Reviewing Policy:**

Affirmed.

**Key Questions For Authors:**

In Table 4 (page 8, line 185) and Figure 4 (line 415), ConDA is compared with PCA and β-VAE. Given that ConDA is designed to integrate with diffusion-based frameworks, could the authors extend their evaluation to include a broader range of generative or representation learning models for comparison?

**Limitations:**

Yes

**Strengths And Weaknesses:**

The paper addresses a critical limitation in diffusion models—the opacity of their latent representations—by proposing a simple yet effective solution: a plug-and-play alignment layer that enhances interpretability without disrupting the original generative process. The idea is both innovative and practical.

The experimental evaluation is thorough, covering five diverse datasets with clearly described setups, metrics, and network architectures. The ablation study is well-reasoned, and the results consistently show that ConDA outperforms existing methods. Overall, the paper makes a strong contribution to improving the controllability and interpretability of diffusion-based generative models.

---

> ### Author Rebuttal · Authors · 2026-03-30
>
> We thank the reviewer for the positive assessment and helpful suggestion.
>
> PCA and $\beta$-VAE were included as matched latent-structuring baselines applied to the same inverted diffusion latents, so that the comparison isolates how the latent organization method affects controllability and reconstruction. In that setting, ConDA consistently improves over both PCA and $\beta$-VAE on test-sequence prediction.
>
> We agree that broader comparisons would be valuable. The current appendix already includes robustness across alternative inversion and diffusion backbones, including Rex-RK4 and IsoDiff (Appendix C.3). To address the reviewer's suggestion more directly, we have also added a direct conditional-control baseline (varying only the conditioning variable during diffusion sampling) in **Table A**. This comparison is complementary to the existing PCA and $\beta$-VAE baselines, since it tests whether varying the conditioning variable alone can provide the same degree of controllable interpolation as ConDA. The results show that it does not, further supporting the value of learning a structured control geometry over diffusion latents.
>
> **Table A: Condition Interpolation Baseline.**
> |Data |   Method          | PSNR ↑               | SSIM ↑           |RMSE ↓ |
> |-----|-------------------|----------------------|------------------|-------|
> |Fluid| Interp. Condition |  29.44 ± 1.30        |0.74 ± 0.11       |6.58   |
> |     | ConDA (Spline)    |  **35.70 ± 0.36**    |**0.94 ± 0.01**   |**0.00**|
> | Ca2+| Interp. Condition |  31.53 ± 1.24        |0.74 ± 0.06       |8.29   |
> |     | ConDA (Spline)    |  **38.58 ± 0.59**    |**0.93 ± 0.01**   |**0.00**|
> |DISFA| Interp. Condition |  32.29 ± 2.89        |0.79 ± 0.09       |6.90   |
> |     | ConDA (Spline)    |  **38.99 ± 0.23**    |**0.96 ± 0.00**   |**0.00**|

---

> > ### Author Rebuttal · Reviewer_17dw · 2026-04-01
> >
> > The authors added another table for complementary support, this maitains the same level of the paper, I recommend for acceptance, but keep the score.

---

> > > ### Author Response · Authors · 2026-04-07
> > >
> > > We thank Reviewer 17dw for their positive assessment and for recommending acceptance. Our added conditional control baseline was intended to directly test whether conditioning interpolation alone can provide the same controllable interpolation as ConDA, and the results support the value of learning a structured control geometry over diffusion latents. We appreciate the suggestion regarding broader comparison classes and will clarify the relationship to additional model families clearly in the revision.

---

### Official Review · Reviewer_rGMX · 2026-03-09

**Soundness:** 1
**Presentation:** 3
**Significance:** 2
**Originality:** 3
**Overall Recommendation:** 2
**Confidence:** 3

**Summary:**

This paper learns an interpretable space given an input from a pre-trained diffusion latent space using contrastive learning. This allows smooth nonlinear traversal for physical and biological applications. The proposed method relies on an inversion method to first turn a latent into a noise from the prior distribution, and use the contrastive encoder to map it into the learned space. After applying the target operations to the features in this space, a kNN Decoder is used to convert it back to the diffusion latent space for the sampling process.

**Compliance With Llm Reviewing Policy:**

Affirmed.

**Key Questions For Authors:**

I don't have other questions.

**Limitations:**

Yes

**Strengths And Weaknesses:**

While designing a structured space that allows interpretable operations such as interpolation is an important problem for many scientific applications, learning the maps between the diffusion latent space and the contrastive embedding space is important and non-trivial.

Although this paper tackles the encoding process (from the diffusion latent space to lower-dimensional space) using a contrastive learning framework, which is reasonable, the decoding process is done with a brittle k nearest neighbor weighting process.  Basically given an interpolated embedding in the low-dimensional space, its k nearest neighbors in lower-dimensional space are selected and weights based on the proximity of these neighbors are applied to their corresponding high-dimensional diffusion latents to reconstruct the diffusion latent. This assumes that the decoding process is linear, which is clearly not the case.

In face, given the extremely low dimensionality of the structured embedding space, such as $d=8$ for fluid dynamics and $d=3$ for other domains like calcium imaging and facial expressions, reconstructing the diffusion latents is really challenging. To demonstrate this, if one applies PCA to the diffusion latent of a natural image and visualize its first three principal components as the RGB channels, the resulting image looks quite similar to the input image. Therefore. decoding a small latent vector with less than 10 channels to a latent image is not simpler than the original image generation problem. To this end, the kNN decoder is suboptimal to address this problem.

---

> ### Author Rebuttal · Authors · 2026-03-30
>
> We thank the reviewer for focusing on the lifting step, which is indeed central to the method.
>
> We agree that ConDA should not be interpreted as learning a globally invertible low-dimensional generative code. That is not our claim. ConDA is intended as a local control manifold: editing is performed in low-dimensional $C$, lifting back to the diffusion latents $Z$ is local and neighborhood-preserving, and high-fidelity rendering is still carried out by the diffusion model in $Z$.
>
> This is precisely why we use kNN lifting rather than a high-capacity global decoder. The purpose of the lifting step is not to reconstruct arbitrary points from a tiny code, but to keep edited points anchored to nearby valid diffusion latents during local interpolation, extrapolation, and trajectory editing.
>
> The kNN lifting step does not assume a global linear mapping from $C$ to $Z$. While each reconstruction is expressed as a weighted sum of neighboring diffusion latents, both the neighbor set and the weights depend on the query point, so the overall mapping is input-dependent and therefore nonlinear. This is better understood as a local manifold approximation than as a global linear projection.
>
> Empirically, we already show this local lifting is effective rather than brittle. **Appendix C.4** shows that **kNN lifting consistently outperforms a tuned MLP decoder** across the reported datasets. **Appendix C.5** also shows little degradation over a broad range of low embedding dimensions, indicating that the method is not critically dependent on a high-dimensional $C$ space. The main results further show that operations performed in ConDA space improve interpolation, extrapolation, and downstream classification relative to operating directly in diffusion latent space.
>
> We have also included a locality ablation by perturbing embeddings in $C$ as $c' = c + \alpha \epsilon$, with $\epsilon \sim \mathcal{N}(0, I)$, and measuring reconstruction quality as a function of edit distance $d_C = \|c' - c\|$. As shown in Table B, reconstruction error (RMSE) increases monotonically with distance across all datasets, while PSNR and SSIM degrade smoothly. This directly addresses the concern that lifting from very low-dimensional $C$ must act as a brittle global decoder, as performance is stable for local edits and degrades gradually as one moves farther off-manifold. This behavior is consistent with the design of the kNN lifting step as a local, geometry-preserving operator rather than a global inverse mapping.
>
> We agree that reconstructing high-dimensional latents from arbitrary low-dimensional vectors is a difficult problem. However, this is not our setting: ConDA does not decode arbitrary points in $C$, but performs local, data-supported lifting near the data manifold using nearby valid diffusion latents.
>
> **Table B: Locality Analysis in ConDA Space.**
> | Data        | Distance | PSNR ↑               | SSIM ↑      | RMSE ↓ |
> |-------------|----------|----------------------|----------------------|------|
> | Fluid       | 0.0      | 35.47 ± 0.53         |0.95 ± 0.01  | 0.00   |
> |             | 0.6      | 35.46 ± 0.53         |0.95 ± 0.01  | 0.68   |
> |             | 1.2      | 35.01 ± 1.65         |0.93 ± 0.07  | 4.50   |
> |             | 1.8      | 34.72 ± 1.89         |0.92 ± 0.08  | 6.54   |
> |             | 2.4      | 34.19 ± 2.47         |0.90 ± 0.11  | 8.09   |
> |             | 3.0      | 33.30 ± 3.03         |0.86 ± 0.14  | 9.75   |
> | Ca2+        | 0.0      | 38.10 ± 0.49         |0.93 ± 0.01  | 0.00   |
> |             | 0.6      | 37.74 ± 0.53         |0.92 ± 0.01  | 8.63   |
> |             | 1.2      | 37.46 ± 0.61         |0.92 ± 0.01  | 12.20  |
> |             | 1.8      | 37.44 ± 0.54         |0.91 ± 0.01  | 14.53  |
> |             | 2.4      | 37.35 ± 0.52         |0.91 ± 0.01  | 14.51  |
> |             | 3.0      | 36.96 ± 1.28         |0.90 ± 0.03  | 15.05  |
> |  DISFA      | 0.0      | 38.24 ± 0.37         |0.96 ± 0.00  | 0.00   |
> |             | 0.6      | 37.75 ± 1.27         |0.96 ± 0.02  | 4.93   |
> |             | 1.2      | 37.10 ± 2.09         |0.94 ± 0.05  | 7.13   |
> |             | 1.8      | 36.44 ± 2.77         |0.92 ± 0.07  | 8.51   |
> |             | 2.4      | 35.79 ± 3.09         |0.90 ± 0.08  | 9.41   |
> |             | 3.0      | 35.51 ± 2.96         |0.90 ± 0.08  | 10.59  |

---

> > ### Author Rebuttal · Reviewer_rGMX · 2026-04-04
> >
> > Thank you for the detailed reply and additional experiments. While I appreciate the explanation about the motivation of using kNN instead of some more powerful decoders, I'm still not fully convinced by the performance of using kNN to map a low-dimensional vector to a diffusion latent. I would be more convinced if a similar experiment using kNN is performed on a more challenging task such as text-to-image generation or even class-conditioned generation on ImageNet. I believe that the datasets used in the paper are relatively simple and normally require only minimal or local changes, and that is why it is hard to determine whether kNN can really achieve the mapping.

---

> > > ### Author Response · Authors · 2026-04-07
> > >
> > > We thank the reviewer for the continued engagement. To directly address the request for a more challenging setting, we added a Stable Diffusion text-to-image locality stress test using pretrained Stable Diffusion on LAION [1,2]. **Table C** shows that kNN lifting remains close to the source latent for small edits in $C$, with quality degrading progressively at larger distances. Thus kNN acts as a local, neighborhood-preserving lifting operator for structured edits, not as a universal decoder from an arbitrary low-dimensional code.
> > >
> > >
> > > **Table C: Locality Analysis in ConDA $C$ space for Stable Diffusion text-to-image generation.**
> > >  | Distance | PSNR ↑        | SSIM ↑             | RMSE ↓ |
> > > |--------------|------------------|----------------------|-------------|
> > > | 0.0           | 49.88 ± 3.22     |1.00 ± 0.00         | 0.00   |
> > > | 0.1           | 49.70 ± 3.88     |0.99 ± 0.10         | 2.99   |
> > > | 0.2           | 45.02 ± 9.85     |0.78 ± 0.41         | 63.18   |
> > > | 0.3           | 41.12 ± 11.40   |0.60 ± 0.48         | 113.64   |
> > > | 0.4           | 35.71 ± 11.23   |0.36 ± 0.46         | 184.98   |
> > > | 0.5           |  35.09 ± 11.05  | 0.33 ± 0.46        | 193.96   |
> > >
> > >
> > > This is consistent with the method's design. kNN lifting does not need to reconstruct arbitrary high-dimensional outputs on its own because it only needs to land near a valid diffusion latent in $Z$, from which the pretrained diffusion model handles rendering. The choice of kNN is also not ad hoc: the contrastive objective explicitly organizes $C$ so that local neighborhood structure aligns with the auxiliary variables, making a neighbor-based decoder a reasonable counterpart to the learned space [3]. Empirically, kNN outperforms a tuned MLP decoder across all five evaluated domains (Appendix C.4).
> > >
> > > To clarify scope: While **Table C** supports local lifting behavior even in text-to-image generation, we do not claim that kNN will be optimal in every generation setting. In highly multimodal regimes with sharp categorical boundaries, reliable lifting may require denser latent coverage, a higher-dimensional $C$, or a more expressive lifting method. However, our target settings  (continuous spatiotemporal scientific systems spanning fluid flow, in vivo neural recordings, clinical neurostimulation, naturalistic facial dynamics, and primate electrophysiology) represent a broad and important class of problems where local trajectory editing is the natural operating regime, and where the evaluated dynamics are neither minimal nor trivially local. The consistent improvements across these five distinct domains support that the approach is effective well beyond a single narrow application.
> > >
> > > **References:**
> > >
> > > [1] Rombach, R., Blattmann, A., Lorenz, D., Esser, P., & Ommer, B. (2022). High-resolution image synthesis with latent diffusion models. In CVPR (pp. 10684-10695).
> > >
> > > [2] Ghosh, D., Hajishirzi, H., & Schmidt, L. (2023). Geneval: An object-focused framework for evaluating text-to-image alignment. NeurIPS, 36, 52132-52152.
> > >
> > > [3] Damrich, S., Böhm, J. N., Hamprecht, F. A., & Kobak, D. (2023). From t-SNE to UMAP with contrastive learning. In ICLR.

---

### Official Review · Reviewer_a1a1 · 2026-03-13

**Soundness:** 3
**Presentation:** 3
**Significance:** 2
**Originality:** 2
**Overall Recommendation:** 4
**Confidence:** 3

**Summary:**

The paper focus on learning a structured latent space for diffusion model to enable better controllability. To learn the control latent space is learned via contrastive loss. Experiments show improvements in alignment quality and generation fidelity across several diffusion tasks.

**Compliance With Llm Reviewing Policy:**

Affirmed.

**Final Justification:**

The novelty of this paper remains a concern to me. But demonstrating the idea that learning a low-dim space enables continuous control in diffusion is still meaningful to some extent.

**Key Questions For Authors:**

1. Compared with classifier-free guidance that steers the sampling process towards the conditioned distribution, what makes this method better?
2. Can one learn a small latent space on the conditions and interpolate the conditions to control the generation process with CFG?
3. How well-behaved are the learned latent spaces? The fact that linear method does not perform as well might indicate that the latent space is not as smooth. Also, can one perform temperature sampling in the latent space as in GAN?

**Limitations:**

Yes

**Strengths And Weaknesses:**

Strengths

1. One of the major issue with diffusion is that the latent space is not as interpolate as GAN, where one can smoothly conduct interpolation to obtain generated results that are meaningful. This paper addresses this issue with diffusion, which can be very useful.

2. The framework proposed by this paper is general and can be applied to all diffusion models. Experiments also covers a wide range of applications and data domain.

Weakness

1. The novelty is limited as it resembles existing contrastive preference learning approaches adapted to diffusion models.
2. The process of obtaining the latent-label pair in diffusion is very expensive by using DDIM inversion.  Also the experiments are relatively short scale, making it unclear how the method performs for large diffusion models trained with a very large dataset.
3. The paper lacks deeper insight into why the contrastive objective is suitable for diffusion alignment, can one find a better loss specifically designed for diffusion models?

---

> ### Author Rebuttal · Authors · 2026-03-30
>
> We thank the reviewer for the thoughtful comments and for highlighting both the importance of controllable diffusion latents and the breadth of the empirical evaluation.
>
> **On novelty (W1)** We agree that the contrastive loss itself is not the novelty. Our contribution is a post-hoc control geometry for pretrained diffusion latents that separates editing in a compact auxiliary-aligned space $C$ from rendering in the original diffusion latent space $Z$. This edit/render separation is what enables nonlinear traversal, extrapolation, and class transfer across five dynamical domains in a unified plug-in framework.
>
> **On inversion cost and scalability (W2)** We agree that inversion cost matters. However, ConDA is not tied to DDIM. **Appendix C.2** shows that an alternative inversion method, Rex-RK4 achieves near-identical reconstruction fidelity to DDIM while using only 8 steps instead of DDIM's 1000-step inversion budget, and **Table 7** already reports the corresponding runtime reduction. **Appendix C.3** further shows consistent behavior across alternative inversion and diffusion backbones, including Rex-RK4 and IsoDiff. These results support that the method is solver-agnostic rather than DDIM-specific. In addition to the models in the main paper, **Appendix C.3** includes results on IsoDiff, supporting that the approach transfers across alternative pretrained diffusion backbones.
>
> **On why a contrastive objective (W3)** Our goal is not to redesign the diffusion objective, but to impose auxiliary-variable-aligned local geometry on pretrained diffusion latents. Positives share the relevant auxiliary variable and negatives do not, so the supervised InfoNCE objective directly encourages nearby points in $C$ to correspond to similar underlying dynamics. That is the structure needed for smooth local traversal and class-conditional editing.
>
> **On CFG (Q1)** CFG and ConDA address different problems. CFG is a sampling-time steering mechanism that pushes generation toward a specified condition. ConDA instead learns a reusable low-dimensional control geometry over pretrained diffusion latents that supports interpolation, extrapolation, and interpretable traversal. In that sense the two are complementary rather than competing.
>
> **On whether one could interpolate conditions (Q2)** We agree this is a fair baseline. Interpolating conditions can work when the conditioning variables fully parameterize the relevant variation, but in our settings the auxiliary variables are only partial descriptors and do not by themselves organize latent trajectories into a smooth control manifold. To directly test the reviewer’s suggested alternative, we have added a condition-interpolation baseline (varying only the conditioning variable during diffusion sampling) in Table A, which shows consistently lower fidelity and higher error across all datasets. These results indicate that condition interpolation alone does not provide a well-structured control space, whereas ConDA enables smooth and accurate trajectory-level control.
>
> **On how well-behaved the learned latent spaces are (Q3a)** We do not claim that the ConDA space is globally linear. Rather, the goal is a smooth local geometry that supports meaningful nonlinear traversal. This is exactly what we observe: linear methods are weaker, while spline, LSTM, and TEX-based operators perform well in $C$, indicating a locally smooth but globally nonlinear manifold. We have added a locality analysis in **Table B** (see response to rGMX) showing that reconstruction quality degrades gradually, rather than collapsing abruptly, as edit distance in $C$ increases. This supports the interpretation of $C$ as a stable local control manifold rather than a brittle global inverse map, and provides direct evidence that the learned space is locally smooth and well-behaved for the intended editing regime.
>
> **On temperature sampling in the latent space (Q3b)** We did not study temperature sampling in $C$ in the current work. In principle, one can inject local perturbations in $C$, but our focus here is deterministic trajectory control rather than stochastic latent sampling. We will clarify this scope in the revision.
>
>
> **Table A: Condition Interpolation Baseline.**
> |Data |   Method          | PSNR ↑               | SSIM ↑           |RMSE ↓ |
> |-----|-------------------|----------------------|------------------|-------|
> |Fluid| Interp. Condition |  29.44 ± 1.30        |0.74 ± 0.11       |6.58   |
> |     | ConDA (Spline)    |  **35.70 ± 0.36**    |**0.94 ± 0.01**   |**0.00**|
> | Ca2+| Interp. Condition |  31.53 ± 1.24        |0.74 ± 0.06       |8.29   |
> |     | ConDA (Spline)    |  **38.58 ± 0.59**    |**0.93 ± 0.01**   |**0.00**|
> |DISFA| Interp. Condition |  32.29 ± 2.89        |0.79 ± 0.09       |6.90   |
> |     | ConDA (Spline)    |  **38.99 ± 0.23**    |**0.96 ± 0.00**   |**0.00**|

---

> > ### Author Rebuttal · Reviewer_a1a1 · 2026-04-04
> >
> > I thank the author for the response and the rebuttal. The novelty part is hard to address as it's inhabit to the approach. But I do think this work could be a meaningful addition to improving the controllability of diffusion models.
> > Therefore I will raise my score.

---

> > > ### Author Response · Authors · 2026-04-07
> > >
> > > Thank you for your thoughtful feedback and for raising the score. To clarify the novelty: the core contribution is the post hoc control geometry itself (separating editing in $C$ from rendering in $Z$) rather than the contrastive loss per se. Existing controllability methods typically steer generation by modifying the model's native latent/state representations, attention/conditioning pathways, or denoising process directly. ConDA instead introduces an external, low-dimensional control space learned on top of pretrained diffusion latents, in which editing and rendering are explicitly decoupled. This enables nonlinear traversal, extrapolation, improved interpretability, and counterfactual or class-transfer editing within a unified plug-in framework validated across five domains (a combination of capabilities not offered by prior controllability approaches we are aware of). We will state these distinctions from prior work more clearly in the revision.

---

### Decision · Program_Chairs · 2026-04-30

**Decision:**

Accept (regular)

**Comment:**

The paper addresses structuring diffusion latent spaces for controllable generation with a general-purpose plug-and-play framework validated across five diverse scientific domains. The core disagreement centers on Reviewer rGMX's concern about the kNN decoder's adequacy, which the authors addressed with new evidence (locality ablation, MLP comparison, Stable Diffusion stress test). Reviewer rGMX's remaining objection is about scope (wanting ImageNet-class validation) rather than a demonstrated failure within the paper's stated scope — the method works consistently well across all evaluated domains. Also, Reviewer a1a1's raised score. Therefore, although the novelty concern is real, I am leaning towards a borderline accept.